

# Non-coplanar magnetism, topological density wave order and emergent symmetry at half-integer filling of moiré Chern bands

**Patrick H. Wilhelm[1*], Thomas C. Lang[1],**
**Mathias S. Scheurer[1] and Andreas M. Läuchli[2,3]**

**1** Institut für Theoretische Physik, Universität Innsbruck, A-6020 Innsbruck, Austria
**2** Laboratory for Theoretical and Computational Physics,
Paul Scherrer Institute, 5232 Villigen, Switzerland
**3** Institute of Physics, École Polytechnique Fédérale de Lausanne (EPFL),
1015 Lausanne, Switzerland

⋆ patrick.wilhelm@uibk.ac.at

## Abstract

Twisted double- and mono-bilayer graphene are graphene-based moiré materials hosting strongly correlated fermions in a gate-tunable conduction band with a topologically non-trivial character. Using unbiased exact diagonalization complemented by unrestricted Hartree-Fock calculations, we find that the strong electron-electron interactions lead to a non-coplanar magnetic state, which has the same symmetries as the tetrahedral antiferromagnet on the triangular lattice and can be thought of as a skyrmion lattice commensurate with the moiré scale, competing with a set of ferromagnetic, topological charge density waves featuring an approximate emergent O(3) symmetry, 'rotating' the different charge density wave states into each other. Direct comparison with exact diagonalization reveals that the ordered phases are accurately described within the unrestricted Hartree-Fock approximation. Exhibiting a finite charge gap and Chern number $|C| = 1$, the formation of charge density wave order which is intimately connected to a skyrmion lattice phase is consistent with recent experiments on these systems.



# 1  Introduction

The interplay of topology and interactions is at the heart of the condensed matter community's ongoing fascination with moiré materials [1–3]. Not only do they provide a platform to engineer band structures of minimal bandwidth, dramatically increasing the importance of electron-electron interactions, but they can also be set up to host topologically non-trivial Chern bands — paving the way for the emergence of novel ordered phases. Twisted double-bilayer graphene (TDBG) [4–6] is composed of four stacked layers of graphene, where the top and bottom pairs are in a Bernal configuration while the middle ones are twisted by a small angle $\theta$, typically around $\theta \simeq 1.3°$, which results in a long-range moiré pattern with a period of about 10 nm. Similarly, twisted mono-bilayer graphene (TMBG) [7] is engineered by slightly twisting a Bernal stack relative to a single graphene sheet, totalling in three graphene layers. As with other 2D moiré systems, the density of electrons may be efficiently tuned via electrostatic doping. However, TDBG and TMBG stand out in the landscape of moiré materials due to their lack of $C_2$ inversion symmetry, which leads to finite Berry curvature in a given valley. It also allows to induce a gap at charge neutrality via an external displacement field and stabilize an energetically isolated conduction band in each valley with Chern number $|C| = 2$.

At an integer number of filled moiré bands, these systems were experimentally found to exhibit a tendency towards polarization of the spin and valley degrees of freedom [4–10]. This is reminiscent of early studies of Chern bands occurring in twisted bilayer graphene [11–13] as well as ABC-trilayer graphene [14,15] aligned with the hexagonal boron nitride substrate, where ferromagnetism as well as an anomalous quantum Hall response was observed. In addition, TDBG was found to support a nesting induced correlated electron-hole state at a slightly higher twist angle, where the carrier concentration could be tuned individually in the top and bottom bilayers [16]. Upon doping away from integer filling, electronic nematic order was observed in TDBG [17]. Moreover, signatures of symmetry broken Chern insulators (SBCIs) with $|C| = 1$ at conduction band filling $\nu = 7/2$ in both TDBG [18] and TMBG [19] highlight the intertwining of topology and translational symmetry breaking order at non-integer filling fractions in these moiré systems. Correlated states at half-integer band fillings are also seen in experiments on related graphene based moiré materials [20–22].

In this paper we numerically study the emergence of correlated phases of strongly interacting fermions at half-integer fillings of a realistic model for the conduction band of TDBG and TMBG. More specifically, we focus on fillings $\nu = 1/2$ as well as $\nu = 7/2$. For this purpose, we use extensive exact diagonalization (ED), fully accounting for quantum fluctuations, complemented by unrestricted Hartree-Fock (HF) calculations, where the ansatz for the HF order parameter is guided by the unbiased ED results.

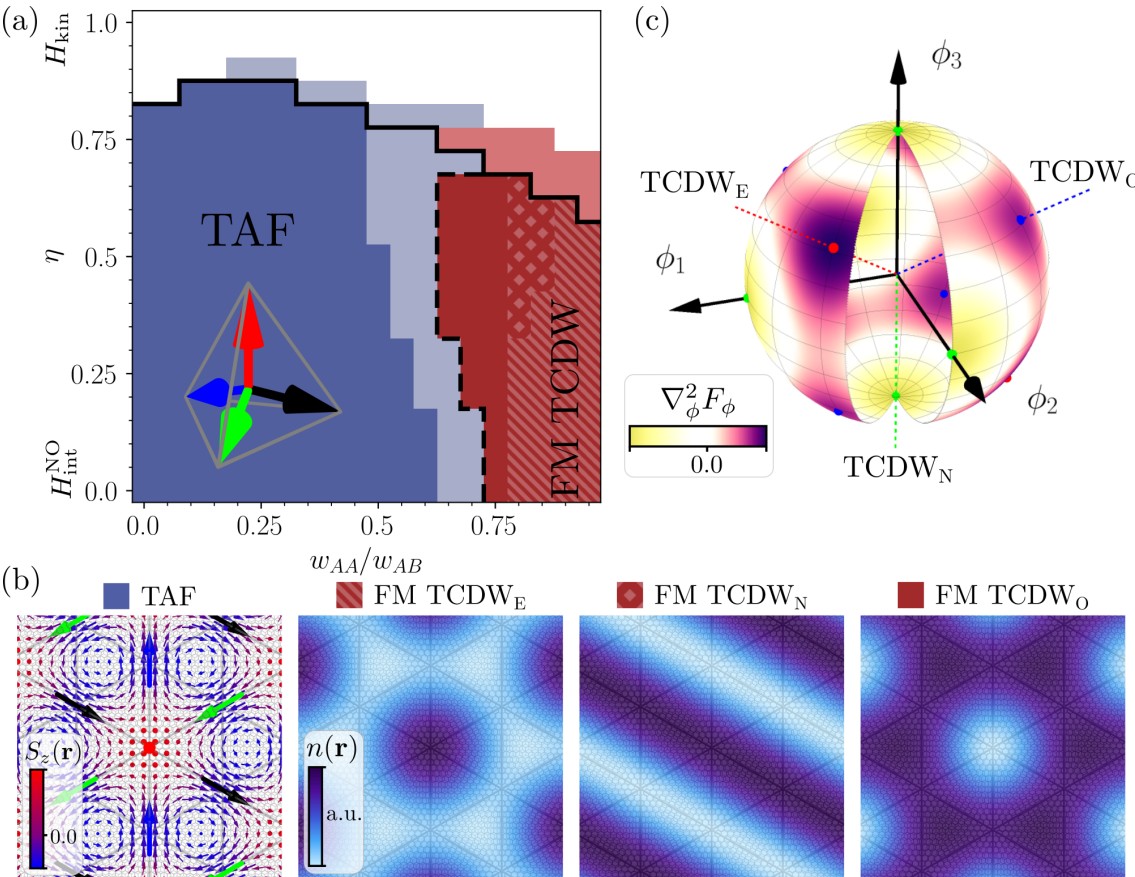

Figure 1: (a) Phase diagram of twisted double-bilayer graphene at conduction band filling $\nu = 1/2$ as a function of the moiré relaxation ratio $w_{AA}/w_{AB}$ as well as $\eta$ controlling the interaction strength, based on the combined data from exact diagonalization and Hartree-Fock. As illustrated in (b), the system realizes a tetrahedral antiferromagnetic (TAF) spin texture in a large parameter regime. At large values of $w_{AA}/w_{AB}$, a ferromagnetic topological charge density wave (TCDW) with an approximate O(3) symmetry emerges. Both types of order break time-reversal symmetry and exhibit a finite, quantized Hall conductance. (c) Illustration of the O(3) TCDW manifold with exemplary anisotropies. Minor cubic and quartic anisotropies select between even (E), odd (O) and nematic (N) realizations of charge order.

We comprehensively demonstrate that the phase diagrams of both systems, exemplarily presented in Fig. 1 (a) for TDBG, are dominated by a tetrahedral antiferromagnetic (TAF) phase, characterized by a non-coplanar spin texture and finite, periodic skyrmion density, which evolves into an almost O(3) degenerate manifold of ferromagnetic, topological charge density waves (TCDWs). The real-space texture of these phases is illustrated in Fig. 1 (b). Related TAF phases were the subject of previous numerical studies in the Mott limit of localized magnetic moments of extended Heisenberg models with a scalar chirality term [23,24] as well as investigations of weak-coupling nesting instabilities on the triangular lattice at the level of mean-field and perturbation theory [25–28]. In a comparable scenario, the phase appeared close to the van Hove energy of a topologically trivial band at relatively weak interactions in an effective moiré-scale Hubbard model for transition metal dichalcogenides [29]. Weak repulsive interactions close to van Hove singularities were also found to promote the emergence of a related topological magnetic texture, called the meron lattice, upon imposing a superlattice potential on the surface of a topological insulator with strong spin-orbit coupling [30]. Con-

versely, we establish the importance of this non-coplanar spin texture to the strong-coupling regime of half-integer filled topological flat bands in the absence of spin-orbit coupling of the continuum model of TDBG and TMBG. Note that, while long-wavelength skyrmions have recently been studied [31–35] as excitations of the correlated insulators in twisted graphene systems with $C_2$ symmetry, due to their potential relevance for pairing in the presence of Dirac cones, we find a moiré-scale skyrmion lattice as a ground state away from integer filling. This highlights that moiré superlattices without $C_2$ symmetry and resultant Berry curvature are ideally suited to stabilize magnetic orders with finite scalar spin chirality. Due to the non-coplanar and likely strongly frustrated nature of the magnetic order, these systems could even host more exotic phases like chiral spin liquids.

In addition, we uncover its potentially intimate relationship with the emergence of a continuous (approximate) symmetry leading to a degenerate manifold of distinct ferromagnetic charge density waves (CDWs). These CDWs can be described by a three-component real order parameter $\boldsymbol{\phi} = (\phi_1, \phi_2, \phi_3)^T$, with $\phi_j$ associated with charge-density modulation with wave vector given by the three moiré **M** points, and the approximate symmetry corresponds to an arbitrary O(3) rotation of these components [cf. Fig. 1 (c)]. The identification of such *hidden symmetries* allows for a more profound understanding of the dominant physics in integer-filled flat bands of twisted bilayer graphene [36–38] and hence may be critical for a coherent picture of correlated phases at fractional fillings of moiré Chern bands as well. It does not only shed new light on the ground state selection as a delicate perturbation to a more general set of charge ordered phases but also leads to an exotic form of (pseudo) Goldstone modes. Our finding of an insulating ferromagnetic, topological charge density wave with Chern number $|C| = 1$ readily explains the experimental observation of SBCI signatures in TMBG of Ref. [19].

## 2 Intertwined non-coplanar magnetism and charge order

Our numerical simulations of TDBG and TMBG are based on their respective continuum model description [39, 40], which we briefly recapitulate in App. B, supplemented by a screened density-density Coulomb interaction. The introduction of the moiré modulation creates an effective triangular lattice, where the new length scale is determined by the moiré period $L^M \simeq 10$ nm. As displayed in red in Fig. 2 (a) for TDBG, near a twist angle of $\theta = 1.3°$ a comparatively flat, spin-degenerate, isolated conduction band may be engineered in each valley by tuning an electric displacement field perpendicular to the graphene layers. For the parameters chosen, this band has a finite Chern number $|C| = 2$ and consequently a topologically non-trivial character. At finite values of the displacement field, the only lattice symmetries unbroken are the moiré translations, as well as $C_3$ rotational symmetry. The resulting Hamiltonian is time-reversal $\mathcal{T}$ invariant since the bands from both valleys $\tau = \pm$ are degenerate but for an inversion of the coordinates. Therefore, the Chern number of the flat bands in each valley are opposite in sign and, hence, cancel out as long as $\mathcal{T}$ is not spontaneously broken.

Finally, the independence of spin $\sigma = \uparrow\downarrow$ and the absence of inter-valley scattering terms yield an overall $U(2)_+ \times U(2)_-$ symmetry in our model, corresponding to the separate conservation of spin and charge within each valley. The fact that the approximately flat conduction band is energetically separated from the rest of the spectrum motivates projecting the model to the red bands in Fig. 2 (a), thus, keeping only one band per spin and valley flavor. Denoting the conduction band fermionic annihilation (creation) operators acting in spin and valley space by $\mathbf{c}_{\mathbf{k}}^{(\dagger)}$, the projected, normal ordered many-body Hamiltonian is given by

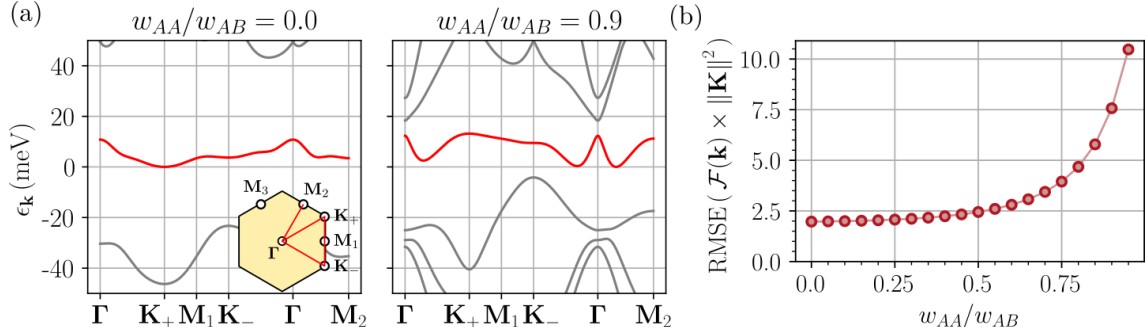

Figure 2: (a) Non-interacting band structure of the ($\tau = -$ valley) TDBG continuum model near the charge neutral Fermi energy and (b) inhomogeneity of the conduction band Berry curvature [defined in Eq. (14)] as a function of the moiré relaxation ratio $w_{AA}/w_{AB}$, see App. B for parameters used. The red bands in (a) have Chern number $|C| = 2$. Their energetic minimum is shifted to zero for illustration purposes.

$$H = \eta \underbrace{\sum_{\mathbf{k}\in\text{MBZ}} \mathbf{c}_{\mathbf{k}}^{\dagger}\epsilon_{\mathbf{k}}\mathbf{c}_{\mathbf{k}}}_{H_{\text{kin}}} + (1-\eta)\underbrace{\frac{1}{2\Omega}\sum_{\mathbf{q}\in\mathbb{R}^2} V(\mathbf{q}):\rho_{-\mathbf{q}}\rho_{\mathbf{q}}:}_{H_{\text{int}}^{\text{NO}}}, \tag{1}$$

where MBZ denotes the Brillouin zone corresponding to the moiré superlattice. The density operators are defined as $\rho_{\mathbf{q}} = \sum_{\mathbf{k}} \mathbf{c}_{\mathbf{k}}^{\dagger}\Lambda(\mathbf{k},\mathbf{q})\mathbf{c}_{\mathbf{k}+\mathbf{q}}$ via the (flavor diagonal) form factor matrix $\Lambda(\mathbf{k},\mathbf{q})$, with entries given by the matrix elements of the corresponding Bloch states of the continuum model; $\Omega$ is the total area of the moiré system and $V(\mathbf{q})$ denotes the Fourier transform of a Yukawa potential. The moiré relaxation ratio $w_{AA}/w_{AB}$ enters the model through the dispersion $\epsilon_{\mathbf{k}}$ as well as the form factors $\Lambda(\mathbf{k},\mathbf{q})$. The convex parameter $\eta \in [0,1]$ allows us to tune the effective interaction strength across the range $[0,\infty]$, while the overall energy scale is maintained.

We start our numerical study by performing ED simulations of ABAB stacked TDBG at a filling fraction $\nu = N_e/N = 1/2$, that is one electron per two moiré unit cells above charge neutrality. The system has a strong tendency towards polarization of fermions into a single valley degree of freedom for a wide range of parameters as shown in Fig. 3 (a), which is consistent with previous HF studies [41] at integer moiré filling that have found spin or valley polarized states to dominate over intervalley coherence. Spontaneous valley polarization breaks time-reversal symmetry $\mathcal{T}$ and consequently introduces a notion of chirality into the system, as the original single-valley moiré band is characterized by a finite Chern number $|C| = 2$. Here, we will focus on this valley polarized (VP), strongly interacting regime at $\eta = 0$. Most interestingly, the remaining SU(2) spin symmetry in a single valley is broken in different ways depending on $w_{AA}/w_{AB}$, in the sense that a phase transition from an antiferromagnet to a maximally polarized ferromagnetic state occurs near $w_{AA}/w_{AB} \simeq 0.7$ in Fig. 3 (b)–(c).

Plotting the filling as a function of the chemical potential in Fig. 3 (d)–(e) for both parameter regimes reveals that, in addition to the expected spin polarized state at $\nu = 1$ [5,6,8,9], $\nu = 1/2$ stands out as a filling fraction with particularly robust insulating signatures. The clear gap in $\mu$ when adding a single additional electron $\delta N_e = 1$ strongly points to an incompressible phase while the classification at other $\nu$ is more ambiguous. In the following sections we discuss what forms of spontaneous symmetry breaking give rise to the incompressible nature of this otherwise itinerant system.

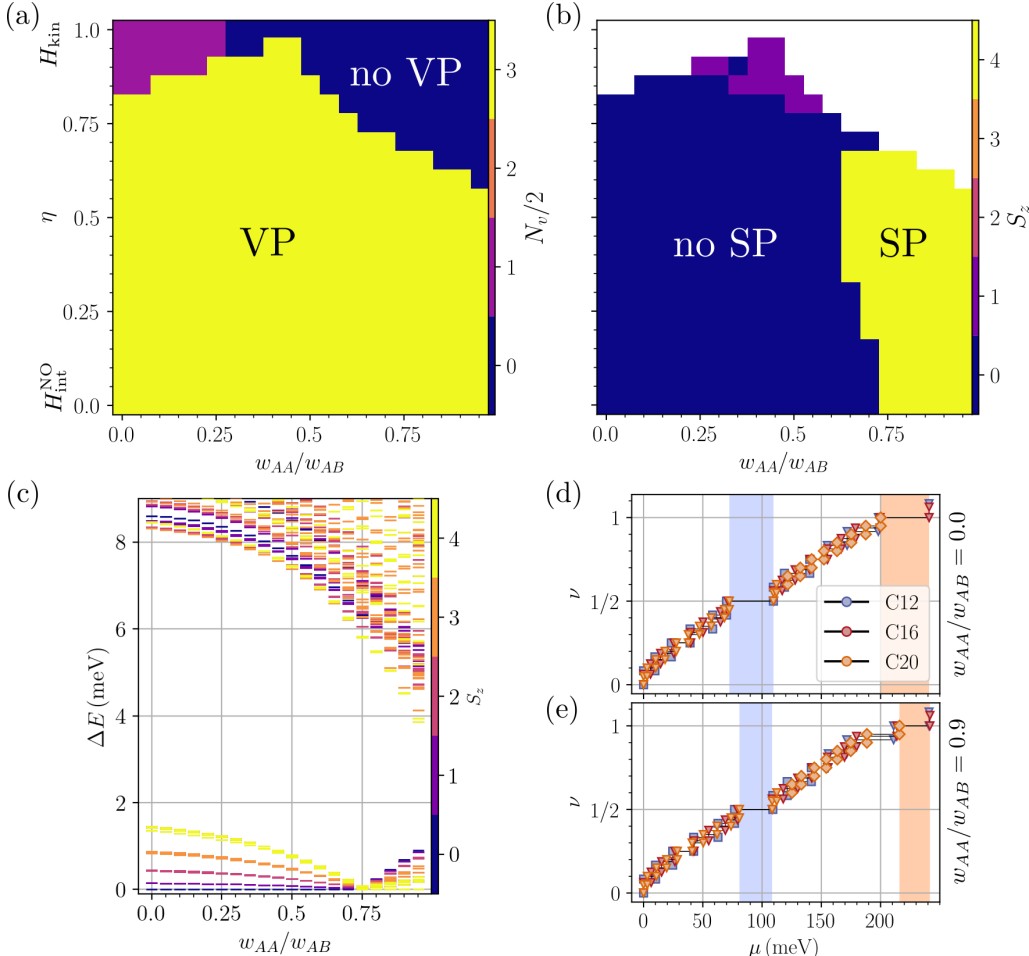

Figure 3: Valley (a) and spin (b) polarization extracted from the lowest energy levels from ED provide evidence for the transition from a time-reversal symmetry breaking singlet (no SP) to a fully spin polarized (SP) state near $w_{AA}/w_{AB} \simeq 0.7$. (c) ED spectrum along the $\eta = 0$ (flat band) slice in (b), corresponding to a pure interaction Hamiltonian. The low lying levels below 2 meV encode the "tower of states", a clear spectral signature for the pattern of continuous symmetry breaking discussed in the main text. (d)–(e) In addition to $\nu = 1$ band filling (orange shaded region), the prominent plateau at $\nu = 1/2$ (blue shaded region) as a function of the calculated chemical potential [defined in Eq. (20)] points to an incompressible, insulating phase at $\nu = 1/2$ for both (d) $w_{AA}/w_{AB} = 0.0$ as well as (e) $w_{AA}/w_{AB} = 0.9$. The used simulation clusters are (a) C12, (b)–(c) C16 and (d)–(e) disclosed in the legend of (d). Square/triangular/diamond shaped markers in (d) and (e) indicate data from spin and valley resolved/valley polarized/spin and valley polarized simulations.

## 2.1 Tetrahedral antiferromagnet

The type of the interaction induced symmetry broken state manifests in the structure of the energy levels as provided by ED. The low-energy spectrum in Fig. 3 (c) near $w_{AA}/w_{AB} \simeq 0$ reveals a set of low lying energy levels forming exact SU(2) spin multiplets with $S = 0, \dots, 4 = N_e/2$, which are well separated by a gap from higher excitations. The momentum-resolved low-energy manifold of states shown in Fig. 4 (a) reveals that all levels are located either at zero center of mass (COM) momentum $\mathbf{k}_{COM} = \mathbf{\Gamma}$ or at the MBZ boundary at half a moiré reciprocal lattice vector $\mathbf{k}_{COM} = \mathbf{M}_j$. Furthermore, the number of approximately degenerate spin multi-

plets increases itself as $2S + 1$, which strongly hints at a non-collinear magnetic symmetry breaking pattern. The energy fine structure of the so called *tower of states* (TOS) has two characteristic features at display here. On the one hand the energy splitting has a $S(S + 1)$ form for fixed size, and a common $1/N$ scaling with the system size [42,43]. Both of these hallmark features are clearly visible in Fig. 4 (b) — a strong indicator for the spontaneous breaking of the continuous SU(2) spin symmetry. Contributions from the distinct momenta $\mathbf{M}_j$ as well as the collapse of the Anderson tower of states are key signatures also found in studies of the extended Heisenberg model on the triangular lattice [24]. This and related work [23] established that along with the chiral spin liquid phase, the TAF formed by localized magnetic moments [depicted in Fig. 1 (b)] is stabilized by the introduction of a scalar spin chirality term $\mathbf{S}_i \cdot (\mathbf{S}_j \times \mathbf{S}_k)$, where the sites $i, j, k$ form a nearest-neighbor triangle in real-space. Such a term can be induced by an external orbital magnetic field, or by the non-trivial topology of the underlying Chern band, as was found by perturbative expansions of the Haldane model at Mott insulator filling [23].

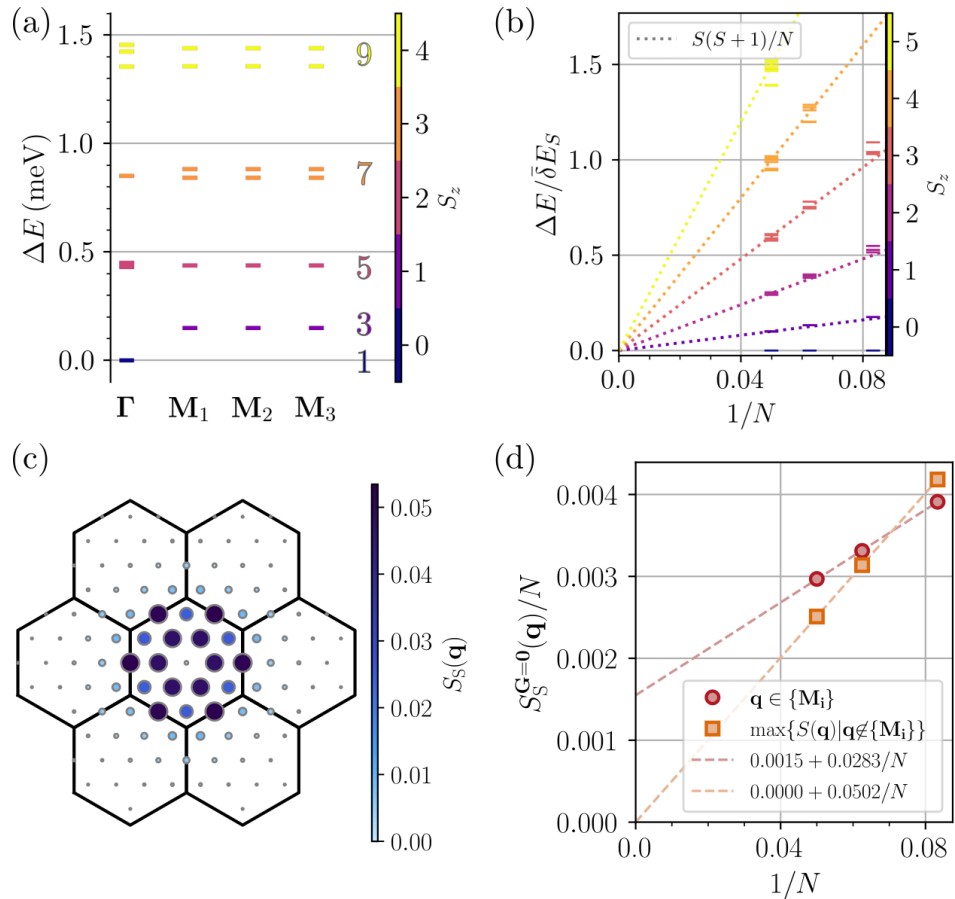

Figure 4: ED evidence for non-coplanar tetrahedral magnetic order at $w_{AA}/w_{AB} = 0$. (a) Multiplets located at center of mass momenta $\mathbf{\Gamma}$ and $\mathbf{M}_j$ form a tower of states characteristic for the tetrahedral antiferromagnet. (b) The low-energy spectrum collapses $\propto S(S + 1)/N$ with increasing system size $N$, a clear manifestation of the spontaneously broken SU(2) symmetry. (c)-(d) Measurements of the spin-$S_z$ structure factor $S_S(\mathbf{q})$ point to the formation of long-range spin order. The peaks at $\mathbf{q} = \mathbf{M}_j$ in (c) (here $N = 16$) extrapolate to finite values in the thermodynamic limit in (d), while other signals vanish. $\mathbf{G} = \mathbf{0}$ indicates momentum transfers $\mathbf{q}$ located in the first moiré Brillouin zone.

The critical importance of band topology at fractional fillings is corroborated by the observation that the $w_{AA}/w_{AB}$-region of TAF stability in Fig. 3 (b)–(c) correlates well with the regime where the Berry curvature $\mathcal{F}(\mathbf{k})$ of the conduction band is homogeneously distributed, see Fig. 2 (b). The k-space homogeneity of the Berry curvature could be of similar importance to the formation of chiral phases in this fractionally filled system as was observed in numerical studies of fractional Chern insulators [44–46].

To fortify our TOS analysis of the low-energy spectrum, we evaluate the signatures of translational symmetry breaking in the many-body ground state $|\Psi_0\rangle$ using the static spin structure factor

$$S_S(\mathbf{q}) = \frac{1}{N}\left\|\left(\tilde{\rho}_{\uparrow,\mathbf{q}} - \tilde{\rho}_{\downarrow,\mathbf{q}}\right)|\Psi_0\rangle\right\|^2, \tag{2}$$

on the topmost layer. Here, the spin density operators are defined as $\tilde{\rho}_{\sigma,\mathbf{q}} = \sum_{\tau,\mathbf{k}} \lambda^\tau(\mathbf{k},\mathbf{q}) c^\dagger_{\sigma,\tau,\mathbf{k}} c_{\sigma,\tau,\mathbf{k}+\mathbf{q}}$, where $\lambda^\tau(\mathbf{k},\mathbf{q}) = \langle P_{A_1} u_\tau(\mathbf{k})|P_{A_1} u_\tau(\mathbf{k}+\mathbf{q})\rangle$ are the form-factors in valley $\tau$ associated with the Bloch states projected onto the dominant $A$ sublattice of the first graphene layer [cf. App. B].

The spin structure factor of the TAF state is shown in Fig. 4 (c), which, as $S_S(\mathbf{q})$ quickly vanishes beyond the first MBZ, reveals that the spin orientations are dominantly evolving over the moiré length scale. The finite size extrapolation in Fig. 4 (d) establishes that only the signals with $\mathbf{q} = \mathbf{M}_j$ survive in the thermodynamic limit (TDL), corresponding to long-range spin order with a $2 \times 2$ unit cell and effective magnetic moments aligned towards the corners of a regular tetrahedron.

Furthermore, the pattern is invariant under the product of $C_3$ and an appropriately chosen spin rotation, such that the charge density remains invariant under moiré translations. As illustrated in the topmost box of Fig. 1 (b), the real-space spin texture of the TAF state can be thought of as a skyrmion lattice. The emergence of a phase with finite spin chirality at half-integer band filling is remarkably reminiscent of earlier works [25–28] on nesting induced multiple-$\mathbf{Q}$ instabilities in Kondo lattice models. However, in stark contrast, the instability in the current situation cannot be caused by nesting of the non-interacting Fermi surface since at $\eta = 0$ the kinetic part of the Hamiltonian vanishes. What is more, in contrast to the Kondo model, the Hamiltonian in Eq. (1) does not incorporate contributions from physically distinct localized magnetic moments but is exclusively composed of dynamic band fermions.

As demonstrated in Fig. 3 (d) at $\nu = 1/2$, the TAF is an insulator, characterized by a finite many-body Chern number $|C| = 1$, a value we explicitly calculate within HF, thus exhibiting an anomalous quantum Hall response. In a mean-field picture, the emergence of a finite Hall conductance can be thought of as the consequence of the coupling between itinerant electrons and spin textures with non-zero spin-chirality which breaks time-reversal symmetry in the charge channel [47].

## 2.2 Emergent, approximate O(3) charge density wave symmetry

Increasing $w_{AA}/w_{AB}$ beyond $\sim 0.7$ in Fig. 3 (c), the order of the TOS levels of the TAF reverses and the ground state becomes ferromagnetic (FM); this can be clearly seen by inspection of the $S_z$ quantum number of the lowest-energy many-body states in Fig. 5 (a). With both spin and valley polarized, the only remaining degree of freedom is charge and any additional symmetry breaking will be associated with charge density wave order. This is indeed what we find as discussed next.

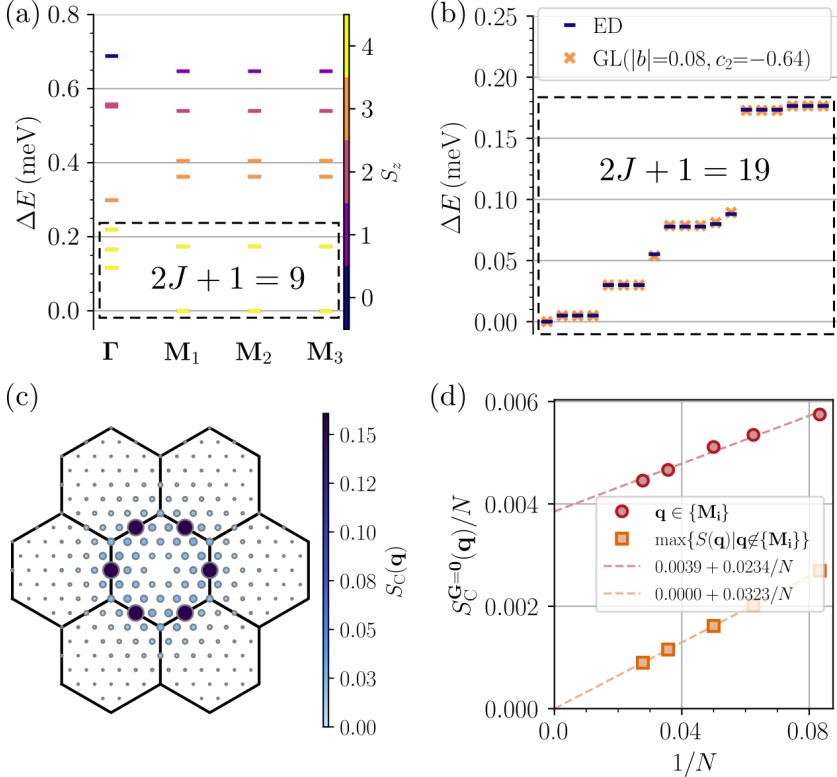

Figure 5: ED evidence for ferromagnetic charge order with an approximate O(3) symmetry at $w_{AA}/w_{AB} = 0.9$. (a) The lowest energy states are completely spin polarized and have center of mass momenta $\mathbf{\Gamma}$ and $\mathbf{M}_j$. (b) The $2J + 1$ lowest levels are nearly degenerate, approximating an O(3) symmetric manifold of states. The relatively minor splitting is well described by an effective O(3) theory with cubic $|b| > 0$ and quartic $c_2 < 0$ anisotropies. (c)-(d) Measurements of the charge structure factor strongly point to the formation of a $\mathbf{q} = \mathbf{M}_j$ charge density wave. The contributions at $\mathbf{q} = \mathbf{0}$ in (c) (here $N = 36$) and (d) are discarded and the values of $b$ and $c_2$ in (b) are given in units of meV. $\mathbf{G} = \mathbf{0}$ indicates momentum transfers $\mathbf{q}$ located in the first moiré Brillouin zone.

Starting from the TAF and increasing $w_{AA}/w_{AB}$, the low-energy spectrum evolves continuously and all low lying levels are still found at the COM momenta $\mathbf{\Gamma}$ and $\mathbf{M}_j$. This fits the presence of pronounced peaks in the charge structure factor $S_C(\mathbf{q})$ at $\mathbf{q} = \mathbf{M}_j$, $j = 1, 2, 3$, in Fig. 5 (c), defined as

$$S_C(\mathbf{q}) = \frac{1}{N} \left\| \tilde{\rho}_{\mathbf{q}} \left| \Psi_0 \right\rangle \right\|^2, \tag{3}$$

where $\tilde{\rho}_{\mathbf{q}} = \sum_\sigma \tilde{\rho}_{\sigma,\mathbf{q}}$ denotes the charge density operator on the first layer. Together with the finite extrapolation to the TDL in Fig. 5 (d), these are hallmark signatures of CDW order. What is unconventional, however, is the amount of ferromagnetic levels that remain bundled up below $\Delta E \lesssim 0.2\,\text{meV}$: a particular pattern of CDW order usually manifests in a fixed number of states corresponding to the ground state degeneracy of the translational symmetry broken states. In contrast, analyzing multiple system sizes, here we find that $2J + 1$ levels are grouped exceptionally close in energy, where $J$ increases linearly with $N$ (or, equivalently, $N_e$ since we work at fixed $\nu$) and equals the maximum total spin $S$ of the ferromagnetic ground state. The *extensive* nature of this suspected ground state manifold is incompatible with a single type of CDW, but instead points at an approximate degeneracy of several density waves, corresponding to an approximate emergent continuous symmetry, as we explain in the following.

To develop intuition for the form and physical meaning of this emergent symmetry, we recall that the spin and valley polarization only leave the spatial modulation of the charge density, $\delta\varrho(\mathbf{r})$, to construct order parameters. Furthermore, the behavior of the charge structure factor found above motivates only taking into account the momenta $\mathbf{M}_j$ in its Fourier expansion,

$$\delta\varrho(\mathbf{r}) = \sum_{j=1}^{3} \phi_j e^{i\mathbf{M}_j\mathbf{r}} + \text{c.c.}\,, \quad \phi_j \in \mathbb{C}. \tag{4}$$

Under translation $T_j$ by a primitive vector $\mathbf{L}_j$, $j = 1, 2, 3$, of the triangular moiré lattice, it holds $\phi_j \to \phi_j$, $\phi_{j' \neq j} \to -\phi_{j' \neq j}$, while $C_3$ acts as $\phi_j \to \phi_{(j+1)\bmod 3}$. The change of the system's free energy $F_\phi$ when turning on CDW order must then have the form $F_\phi \sim \sum_j (a_0 |\phi_j|^2 + \text{Re}[c\,\phi_j^2])$ with $a_0 \in \mathbb{R}$, $c \in \mathbb{C}$ up to quadratic order in $\phi_j$. Consequently, the complex phase of all three $\phi_j^2$ has to be the same, as long as this Ginzburg-Landau (GL) expansion is valid. Since the representation of $T_j$ and $C_3$ both commute with $\phi_j \to e^{i\varphi}\phi_j$, we can set $\phi_j \in \mathbb{R}$ in Eq. (4) without loss of generality. Physically, this means that although $\phi_j \to e^{i\varphi}\phi_j$ generally changes the spatial texture in $\delta\varrho(\mathbf{r})$, its behavior under all symmetries of the system remains the same and, hence, belongs to the same phase. We note that this would be different for twisted bilayer or trilayer graphene as long as their $C_2\mathcal{T}$ symmetry is preserved.

Using real $\phi_j$ and defining $\boldsymbol{\phi} = (\phi_1, \phi_2, \phi_3)^T$, the Ginzburg-Landau expansion becomes

$$F_\phi \sim a\,\boldsymbol{\phi}^2 + b\,\phi_1\phi_2\phi_3 + c_1\left(\boldsymbol{\phi}^2\right)^2 + c_2 \sum_{j<j'} \left(\phi_j\phi_{j'}\right)^2, \tag{5}$$

up to quartic order, where $a = a_0 - |c|$, $b$, $c_1$, $c_2$ are real-valued parameters. As can be shown by straightforward minimization or seen in Fig. 1 (c), where the curvature of $F_\phi$ is shown on a sphere with fixed $\|\boldsymbol{\phi}\|$, there are three different sets of symmetry-unrelated minima: positive $c_2$ favors a state where only one of $\phi_j$ is non-zero. This state breaks $C_3$ symmetry, which is why we refer to it as *nematic*, and doubles the unit cell [cf. the box labeled $\text{TCDW}_\text{N}$ in Fig. 1 (b)]. Negative $c_2$ instead favors configurations where $|\phi_1| = |\phi_2| = |\phi_3| \neq 0$. Although they have the same symmetries, we distinguish states with $\phi_1\phi_2\phi_3$ positive and negative, favored by $b < 0$ and $b > 0$, respectively, since they are not related by symmetry. We refer to these states as $\text{TCDW}_\text{E}$ and $\text{TCDW}_\text{O}$. As can be seen in their respective boxes in Fig. 1 (b), both of these CDW states enlarge the moiré unit cell by a factor of four.

Coming back to the emergent symmetry, we see that $F_\phi$ in Eq. (5) only has discrete symmetries as long as $b$ or $c_2$ is non-zero. If, however, both $b$ and $c_2$ are zero (small compared to $\sqrt{|ac_1|}$ and $c_1$, respectively), we obtain an emergent (approximate) O(3) symmetry; it acts as $\boldsymbol{\phi} \to O\boldsymbol{\phi}$, $O \in$ O(3), i.e., it mixes the three distinct CDW states $\text{TCDW}_\text{N}$, $\text{TCDW}_\text{E}$, and $\text{TCDW}_\text{O}$ defined above. On this note, it should be emphasized that this continuous O(3) symmetry is independent of the $\text{U}(2)_+ \times \text{U}(2)_-$ symmetry of the model, but is an emergent feature in the spin and valley polarized regime of the studied many-body Hamiltonian. To corroborate this Ginzburg-Landau picture and also resolve the finite breaking of the O(3) symmetry within our ED spectra, we note that the approximate symmetry implies that the low-energy space of the many-body Hamiltonian can be described as an effective *macroscopic* angular-momentum operator $\hat{J}$. If the O(3) symmetry were exact, the ground state manifold would just be given by $2J + 1$ exactly degenerate states, where $\hat{J}^2 = J(J+1)\mathbb{1}$. Finite $b, c_2 \neq 0$, however, induces anisotropy terms in the effective Hamiltonian, which lead to the contribution

$$\Delta H_J = \frac{b}{6J^3} \sum_{\substack{(\alpha,\beta,\gamma)\in \\ \pi(x,y,z)}} J_\alpha J_\beta J_\gamma + \frac{c_2}{6J^4} \sum_{\substack{(i,j)\in \\ \{(x,y),(y,z),(x,z)\}}} \sum_{\substack{(\alpha,\beta,\gamma,\delta)\in \\ \pi(i,i,j,j)}} J_\alpha J_\beta J_\gamma J_\delta\,, \tag{6}$$

in analogy to Eq. (5). Here $\pi$ denotes all possible permutations and the normalization by $J^n$ is added in order for the parameters $b$ and $c_2$ to be of similar magnitude across multiple system

sizes. Diagonalization of $\Delta H_J$ leads to $2J + 1$ eigenvalues and fitting to the low-energy states of ED, allows to determine $b$ and $c_2$.

In Fig. 5 (b) we directly compare the energy levels and their degeneracy of the effective model Eq. (6) with the low-energy structure from ED of the full TDBG model for an optimal choice for the parameters $b$ and $c_2$. We find a remarkable agreement with the GL theory, which nearly perfectly reproduces the whole $2J + 1$-dimensional spectrum. Since the spectrum of Eq. (6) is symmetric under $b \to -b$, its sign cannot be inferred from the optimization procedure. In contrast, $c_2$ is found to be necessarily negative for all larger system sizes, and thus energetically favors TCDW$_{\text{E}}$ or TCDW$_{\text{O}}$ over TCDW$_{\text{N}}$.

The analysis of the system's compressibility at $w_{AA}/w_{AB} = 0.9$ in Fig. 3 (e) indicates insulating behavior of the TCDW phase at band filling $v = 1/2$, which is in accordance with charge order.

## 3  Correspondence with Hartree-Fock Slater determinants

HF and generic mean-field calculations are routinely applied in studies of moiré materials [29, 35, 38, 41, 48–50]. Despite making the quite restrictive assumption that the many-body ground state wave function is represented by a single Slater determinant, in certain cases of an integer number of filled bands the method was found to capture the physics of twisted bilayer graphene with surprising accuracy [51, 52]; for twisted bilayer [38, 53, 54] and trilayer [35] graphene, this can be understood analytically by constructing exactly solvable limits where the exact symmetry-breaking ground states are product states. For TDBG, TMBG and for fractional filling studied here, the accuracy of HF has not been tested with unbiased numerics and no realistic exactly solvable limits are known. In order to address this and complement our unbiased, but size-limited, ED results for the current situation at fractional filling, we perform self-consistent unrestricted HF calculations for the band-projected Hamiltonian of Eq. (1). Based on indications provided by the ED many-body spectrum, we allow the HF correlation matrix to simultaneously support translational symmetry breaking in all flavor channels for up to three independent momenta $\mathbf{q}$

$$P_{\sigma,\sigma'}^{\tau,\tau'}(\mathbf{k}, \mathbf{q}) = \langle c_{\sigma,\tau,\mathbf{k}}^{\dagger} c_{\sigma',\tau',\mathbf{k}+\mathbf{q}} \rangle, \tag{7}$$

with $\mathbf{q} \in \{\mathbf{0}, \mathbf{M}_1, \mathbf{M}_2, \mathbf{M}_3\}$. The ordering tendencies of the self-consistent solutions are probed via the following set of observables: The valley polarization $p_v$ and magnetization $m$ are defined as

$$p_v = \frac{1}{N_e}\left|\sum_{\mathbf{k}}\langle \mathbf{c}_{\mathbf{k}}^{\dagger}\sigma_0\tau_z\mathbf{c}_{\mathbf{k}}\rangle\right|, \quad m = \frac{1}{N_e}\left\|\sum_{\mathbf{k}}\langle \mathbf{c}_{\mathbf{k}}^{\dagger}(\sigma_x, \sigma_y, \sigma_z)\tau_0\mathbf{c}_{\mathbf{k}}\rangle\right\|, \tag{8}$$

with $\sigma_\alpha$ ($\tau_\alpha$) denoting Pauli matrices in spin (valley) space. Translational symmetry breaking is captured by

$$\mathcal{M}_j^\alpha = \frac{1}{2}\sum_{\tau,\mathbf{k}}\sum_{\xi=\pm}\lambda^\tau(\mathbf{k}, \xi\mathbf{M}_j)\langle \mathbf{c}_{\tau,\mathbf{k}}^{\dagger}\sigma_\alpha\mathbf{c}_{\tau,\mathbf{k}+\mathbf{M}_j}\rangle, \tag{9}$$

at the momenta $\mathbf{q} = \mathbf{M}_j$, $j = 1, 2, 3$, which yields the spin density wave vector $\boldsymbol{\mu}_j = \left(\mathcal{M}_j^x, \mathcal{M}_j^y, \mathcal{M}_j^z\right)$ and the CDW amplitude $\phi_j = \mathcal{M}_j^0$ [the average over $\xi$ compensates

for different contributions of the form factors $\lambda^\tau(\mathbf{k}, \pm\mathbf{M}_j)$]. A characteristic quantity of the TAF phase, sensitive to its chiral, non-coplanar nature, is given by the momentum space scalar chirality $\chi = \boldsymbol{\mu}_1 \cdot (\boldsymbol{\mu}_2 \times \boldsymbol{\mu}_3)$. Being odd under $\mathcal{T}$, but invariant under $C_3$ and spin rotations, it is reminiscent of the scalar spin chirality term induced by an orbital magnetic field in lattice Heisenberg models [55].

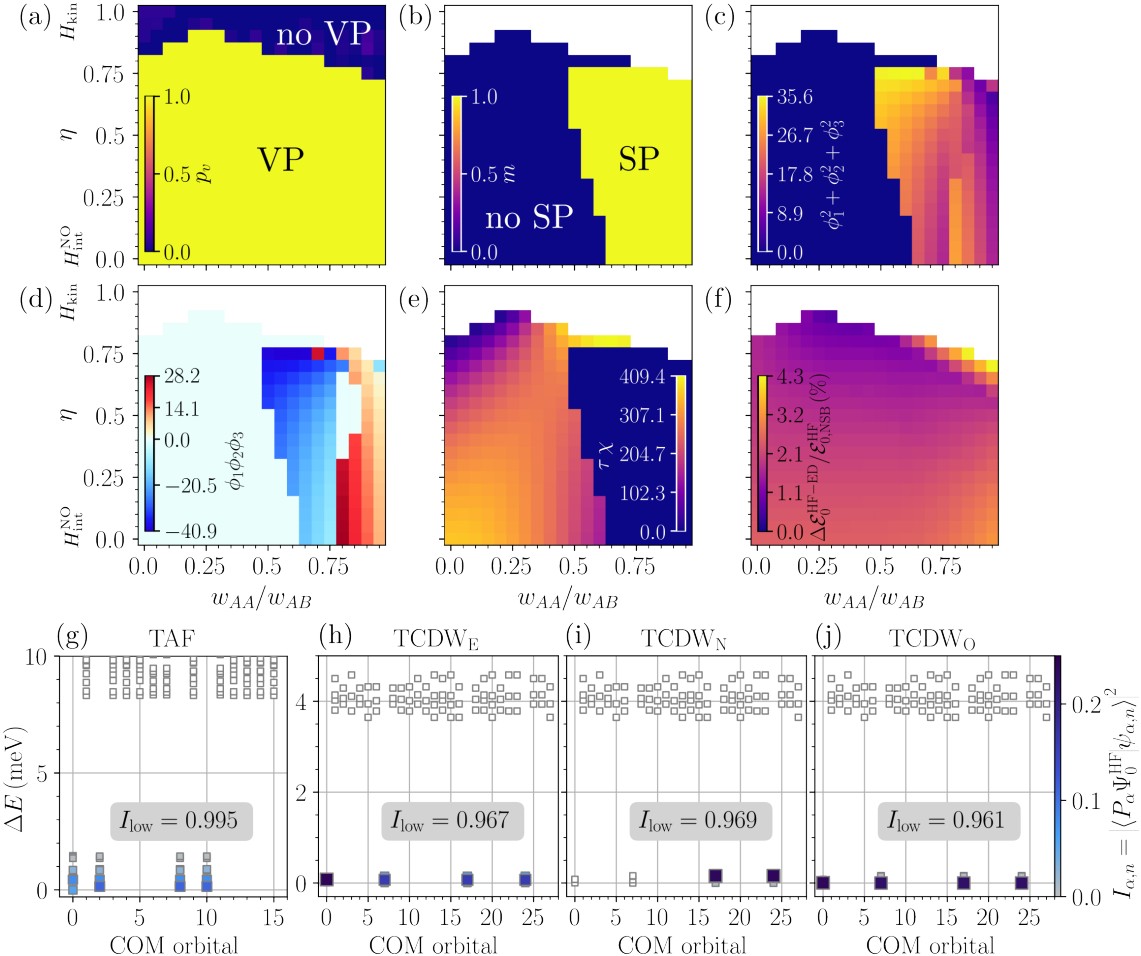

Figure 6: (a)-(b) The relative valley polarization $p_v$ and magnetization $m$ from HF reproduce the respective quantities from Fig. 3 (a)–(b) to a high degree of accuracy. (c) The finite charge order parameters $\phi_j$ suggest general charge density wave tendencies in the ferromagnetic regime while (d) the product $\phi_1\phi_2\phi_3$ discriminates TCDW$_E$ ($>0$), TCDW$_O$ ($<0$) and the $C_3$ breaking TCDW$_N$ ($\simeq 0$) order. (e) Measurements of the scalar chirality $\chi$ confirm the non-coplanar nature of the antiferromagnet. (f) The small difference in ground state energies per site relative to the non-symmetry-breaking state, attributes a surprising quality to the HF solution. (g)–(j) The HF Slater determinants are almost completely composed of ED states contained in the ground state manifolds at COM orbitals $\boldsymbol{\Gamma}$ and $\mathbf{M}_j$. In (a)–(c), (d)–(e) we used cluster C144, (f) is combined from C144 in HF and C12 in ED, (g) is computed on C16 and (h)–(j) is C28.

In agreement with the previous results from ED in Fig. 3, the HF procedure yields full valley polarization $p_v = 1$ for most of the studied phase diagram in Fig. 6 (a), except for $\eta$ close to 1, where the kinetic term in Eq. (1) dominates. Near $w_{AA}/w_{AB} \simeq 0.6$ (at $\eta = 0$) in Fig. 6 (b), the system transitions from a state with zero magnetization ($m = 0$; labeled by "no SP") to a ferromagnetically ordered one ($m = 1$; indicated by "SP")—again in qualitative agreement with

ED. In contrast to the $m = 0$ phase, the completely polarized state exhibits pronounced CDW order $\sum_j \phi_j^2 > 0$ in its entire region of stability [cf. Fig. 6 (c)]. A cubic combination of the real CDW amplitudes $\phi_1 \phi_2 \phi_3$ in Fig. 6 (d) subdivides the SP regime into $C_3$ symmetric $TCDW_E$ and $TCDW_O$ domains partially separated by a distinct nematic $TCDW_N$ phase. The entire region with $m = 0$ exhibits finite and non-coplanar $\boldsymbol{\mu}_j$ at wavevectors $\mathbf{q} = \mathbf{M}_j$, which results in finite values of $\chi$ in Fig. 6 (e), whose sign is governed by the spontaneously polarized valley $\tau$. The same holds true for the composite Chern number of the HF state, which we explicitly calculate to be $\tau C = 1$ for the TAF as well as all TCDWs. A direct comparison of the ground state energies per site $\Delta \mathcal{E}_0^{HF-ED} = \mathcal{E}_0^{HF} - \mathcal{E}_0^{ED}$ relative to the energy of the non-symmetry breaking state $\mathcal{E}_{0,NSB}^{HF}$ in Fig. 6 (f) affirms the quality of the HF solution on a quantitative level. With no more than $\sim 4.3\%$ deviation from the per-site energy provided by ED, the optimized Slater determinant comes remarkably close to the true ground state energy on the finite lattice. Fig. 6 (g)–(j) reiterate on the connection of the self-consistent Slater determinants with the ED states by explicitly representing them in the Fock state basis and subsequently performing a decomposition in the ED spectrum. Both the TAF as well as the TCDW phases reach overlaps of more than 96% upon accumulating the contributions from all symmetry sectors in the ground state manifolds. What is more, the finite order parameter of the devised phases opens an interaction induced gap at the Fermi level, which perfectly agrees with the ED results in Fig. 3 (d)–(e) and underlines the bulk insulating nature of the obtained phases. The combination of our ED and HF results culminates in the phase diagram presented in Fig. 1 (a).

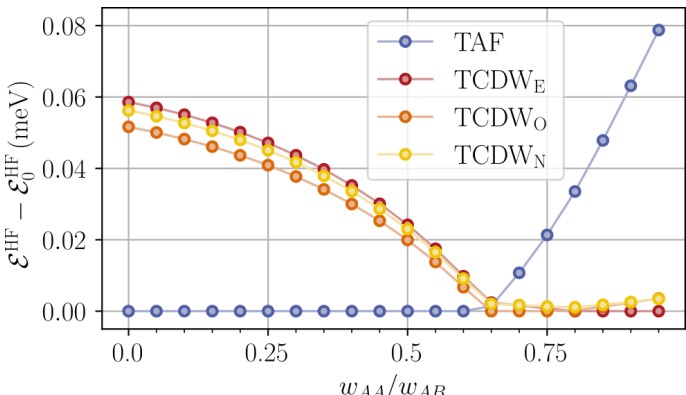

Figure 7: Energies per site of the TAF and TCDW phases relative to the HF ground state as a function of $w_{AA}/w_{AB}$. The different forms of TCDW order stay close in energy, especially for $w_{AA}/w_{AB} \gtrsim 0.7$, while there is a significant gap to the TAF phase. The crossover of the TAF and TCDW energy lines is reminiscent of the way the spin polarized levels and the singlet evolve through the phase transition in Fig. 3 (c). The applied discretization is C576.

Restricting the correlation matrix in Eq. (7) to each of the subspaces corresponding to the four discussed self-consistent solutions, TAF and $TCDW_{E,O,N}$, we may track the evolution of their relative energies. Fig. 7 clearly shows the crossover of the TAF and the set of ferromagnetic TCDW phases near $w_{AA}/w_{AB} \simeq 0.6$–$0.7$. Most notably, whilst there is a clear energetic separation of the TAF and TCDW phases (except for the region of the phase transition) the TCDW phases remain very close in energy throughout the tuning range, especially for $w_{AA}/w_{AB} \gtrsim 0.7$. This is in accordance with the previously established spectral evidence for the emergence of an approximate O(3) symmetry in the CDW channel. In general, the overall shape of the crossover in Fig. 7 is reminiscent of how the singlet and maximally polarized multiplets evolve through the spin phase transition in Fig. 3 (c).

As discussed in App. A.2, the states apparently inherit a lot of their quantum geometry from

the original moiré band. Despite the very different nature of the TAF and TCDW phases, their Berry curvatures are very similar. Along with the evolution of the $2J + 1$ FM levels containing the TCDWs from the TAF tower of states, while remaining separated from higher excited ones, [cf. Fig. 3 (c)] this might point to a more intimate connection of antiferromagnetic skyrmion textures and generic topological charge density waves rooted in the topologically non-trivial nature of the moiré Chern band.

# 4 Phase diagrams at $\nu = 7/2$ and connection to experiments

In order to make the direct connection to recent experiments on TDBG and TMBG, we hereby study both systems at filling $\nu = 7/2$ of the $|C| = 2$ spin and valley degenerate conduction bands, which corresponds to a hole filling of $\nu_h = 1/2$. Motivated by insights about the band-projected interaction Hamiltonian in the context of twisted bilayer graphene [38,54] and in order to counteract the large asymmetry of the electron and hole side in the interaction part of Eq. (1), we replace the density operators $\rho_\mathbf{q}$ by $\delta\rho_\mathbf{q} = \rho_\mathbf{q} - \frac{1}{2}\sum_\mathbf{k} \mathrm{Tr}\left[\Lambda(\mathbf{k},\mathbf{q})\right]\delta_{\mathbf{q}\in\mathrm{RL}}$, where $\delta_{\mathbf{q}\in\mathrm{RL}}$ is only non-zero if $\mathbf{q}$ is a reciprocal moiré lattice vector. Whilst omitting normal ordering, this results in a (up to a constant) particle-hole symmetric interaction Hamiltonian

$$H_{\mathrm{int}}^{\mathrm{PH}} = \frac{1}{2\Omega}\sum_{\mathbf{q}\in\mathbb{R}^2} V(\mathbf{q})\delta\rho_{-\mathbf{q}}\delta\rho_\mathbf{q}. \tag{10}$$

The Hamiltonian of Eq. (10) differs from Eq. (1) by a quadratic term which corresponds to a single-band version of the subtraction method applied in Ref. [38] and Ref. [54].

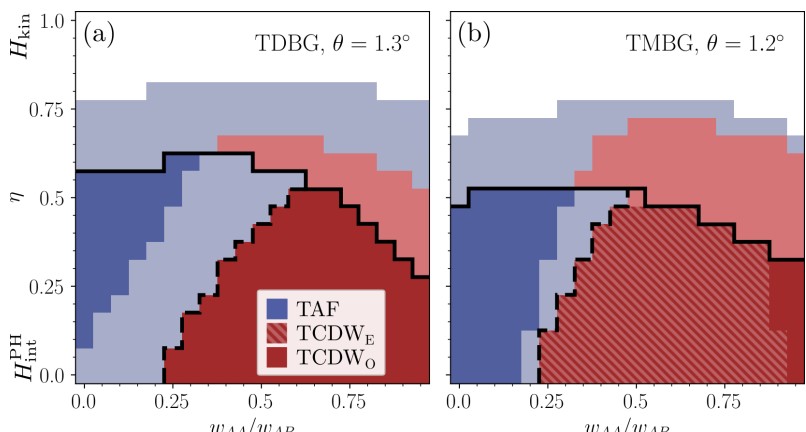

Figure 8: Phase diagram at $\nu = 7/2$ with particle-hole symmetric interactions. Except for the lack of stabilization of the nematic TCDW as the absolute ground state for (a) TDBG as well as (b) TMBG, the same phases as in Fig. 1 (a) compete in the valley polarized parameter regime. Below the solid black line, signatures of valley polarized TAF or TCDW phases are present in ED while the dashed line indicates the spin phase transition. The magnetic order in light-colored regions manifests only in one of the two methods.

Remarkably, antiferromagnetic and ferromagnetic domains in Fig. 8 are located in approximately the same regions of the parameter space for both systems. The TAF is again stable at low $w_{AA}/w_{AB}$, as characterized by its Anderson TOS, whereas $w_{AA}/w_{AB} \gtrsim 0.25$ leads to a spin-polarized TCDW ground state at $\eta = 0$. Due to the particle-hole symmetry of the Hamiltonian $H_{\mathrm{int}}^{\mathrm{PH}}$ at $\eta = 0$, the phase diagram in the purely interacting limit is the same for $\nu = 7/2$

and $\nu = 1/2$. Hence, the shift of the phase transition from $w_{AA}/w_{AB} \simeq 0.7$ in Fig. 1 (a) to $w_{AA}/w_{AB} \simeq 0.2$ in Fig. 8 (a) must be caused by the additional dispersive term only. This might indicate, that nesting processes related to the ones discussed in Refs. [25–28] could play a role in the stabilization of the TAF and TCDW phases, where the effective dispersion stems from the originally normal ordered and band-projected interactions in $H_{int}^{NO}$ in Eq. (1) rather than the usual hopping of electrons.

In the FM regime, the spectrum again exhibits the $2J + 1$-fold quasi-degeneracy rooted in an emergent O(3) CDW symmetry – reproducing the crucial physical features of the $\nu = 1/2$ case. More so, the splitting of the lowest ferromagnetic levels is found to be significantly decreased in relation to the multiplet gap by switching to $H_{int}^{PH}$ and especially the spectrum of TMBG suggests extraordinarily low coefficients for the primary anisotropic terms in the GL theory of $|b| = 0.06\,\text{meV}$ and $c_2 = -0.05\,\text{meV}$ [cf. App. A.1]. Tuning $\eta > 0$, the TAF regime exhibits a more or less pronounced cone-like front of stabilization until the lowest energy state in ED is no longer valley polarized near $\eta \simeq 0.5$.

An explicit calculation of the HF Chern number $|C| = 1$ again attributes a topologically non-trivial character to the wave function. In combination with the insulating, ferromagnetic nature of the TCDW in the realistic parameter range of $w_{AA}/w_{AB} \simeq 0.6$–0.9, our findings are in excellent agreement with the experimental signatures of a ferromagnetic symmetry broken Chern insulator with $|C| = 1$ reported in Ref. [19] for TMBG. The proposed form of translational symmetry breaking along one moiré lattice vector coincides with our definition of the $\text{TCDW}_N$ phase, where strain or boundary effects may play a crucial role in the sub-selection of the ground state from the quasi O(3) degenerate manifold of ordered phases. On the other hand, the absence of SBCI signatures for ABAB stacked TDBG in Ref. [18] may be rooted in the delicate interplay of band-projected interactions and kinetic dispersion contained in Fig. 8 (a). The exact location and shape of the valley polarization transition line may sensitively depend on various system parameters, potentially causing time-reversal symmetry to be restored and any quantum Hall signatures to vanish. Nevertheless, the extraordinary similarity of Fig. 8 (a) and Fig. 8 (b) in combination with the experimental finding that TMBG behaves similar to TDBG at integer fillings [10] holds potential that ABAB stacked TDBG may also host ferromagnetic topological charge density waves at half-integer filling.

## 5 Conclusion

We performed extensive numerical studies of the ground state order in a continuum model description of twisted double-bilayer graphene and twisted mono-bilayer graphene at half-integer filling of the flat Chern conduction band using a combination of exact diagonalization and unrestricted Hartree-Fock methods. We provided conclusive evidence for the emergence of a non-coplanar magnetic state with the symmetries of the tetrahedral antiferromagnet of the underlying triangular lattice and finite skyrmion density in its four-moiré-site unit cell, see upper panel in Fig. 1 (b). While this state is found to be energetically favored for small $w_{AA}/w_{AB}$, a set of almost degenerate spin-polarized charge density wave phases [lower three panels in Fig. 1 (b)] dominates for larger $w_{AA}/w_{AB}$, corresponding to an approximate emergent O(3) symmetry. All of these phases are incompressible and exhibit a finite Chern number. The intimate connection between the phases on either side of this transition is clearly visible in our many-body spectra, where the tower of states of the tetrahedral state evolve into the set of ferromagnetic charge density levels. Remarkably, the explicit Fock space construction of the Hartree-Fock state and its decomposition in the ED ground state manifold establishes that all observed phases are exceptionally close to a fermionic product state.

Our findings provide a natural explanation of the experimental signatures of a ferromag-

netic Chern insulator observed in twisted mono-bilayer graphene [19] in an unbiased way and give perspective on the possibly delicate interplay of interactions and band dispersion responsible for the formation of such phases. Our work further illustrates that graphene moiré systems where $C_2$ symmetry is explicitly broken by the lattice and can thus exhibit bands with finite Berry curvature and Chern numbers in each valley are promising platforms for stabilizing frustrated magnetic phases or potentially even spin-liquid states at fractional filling.

As the tetrahedral antiferromagnet at small $w_{AA}/w_{AB}$ is invariant under a combination of spin rotation and translations, it will not give rise to a charge modulation on the triangular moiré lattice. However, applying a magnetic field, would induce charge modulations that could for instance be observed in future scanning tunneling microscopy experiments [17, 56, 57]. Another perspective for future experiments [58–60] would be to identify the (almost) gapless collective modes associated with the (approximate) emergent O(3) symmetry and its explicit breaking by unintentional or controlled strain (e.g., via a piezoelectric substrate).

The work also poses many important open theoretical questions such as studying the physical consequences of charge-density fluctuations close to the O(3) symmetric limit [61]. We also believe that the emergent symmetry and the proximity of the ground states to Slater determinants we establish could pave the way for an analytical understanding of the strongly correlated low-temperature physics in twisted double- and mono-bilayer graphene.

*Note added:* In the final steps of preparation of this manuscript, experimental signatures of an anomalous Hall effect in ABAB stacked TDBG near $\nu = 7/2$ and its stability under both in- and out-of-plane magnetic fields were reported in Ref. [62]. This is consistent with our results for TDBG and strengthens the established similarity of TDBG and TMBG in that it is in line with the observation of symmetry broken Chern insulators at $\nu = 7/2$ in Ref. [19].

## Acknowledgements

We thank Y. Kwan for valuable discussions on twisted bilayer graphene.

**Funding information**   This work was supported by the Austrian Science Fund FWF within the DK-ALM (W1259-N27). The computational results presented have been achieved in part using the Vienna Scientific Cluster (VSC). M.S.S. acknowledges funding by the European Union (ERC-2021-STG, Project 101040651—SuperCorr). Views and opinions expressed are however those of the authors only and do not necessarily reflect those of the European Union or the European Research Council Executive Agency. Neither the European Union nor the granting authority can be held responsible for them.

## A   Supplementary data

### A.1   Additional many-body spectra

In order to corroborate our conclusions from ED, we hereby present more evidence from many-body spectra. Fig. 9 reassures the persistence of the TOS structure for ABAB stacked TDBG at $\nu = 1/2$ with normal ordered interactions also on a larger, yet less symmetric lattice. The characteristic TAF level structure at $w_{AA}/w_{AB} = 0$ in Fig. 9 (a) again evolves into the imminent quasi-degeneracy of $2J+1 = 11$ maximally spin polarized levels at $w_{AA}/w_{AB} = 0.9$ in Fig. 9 (b).

The introduction of the additional quadratic Coulomb term resulting in a particle-hole symmetric interaction Hamiltonian apparently suppresses the TAF phase at $w_{AA}/w_{AB} = 0$, $\eta = 0$ such that all low-energy multiplets are almost completely degenerate in Fig. 9 (c). In fact, in

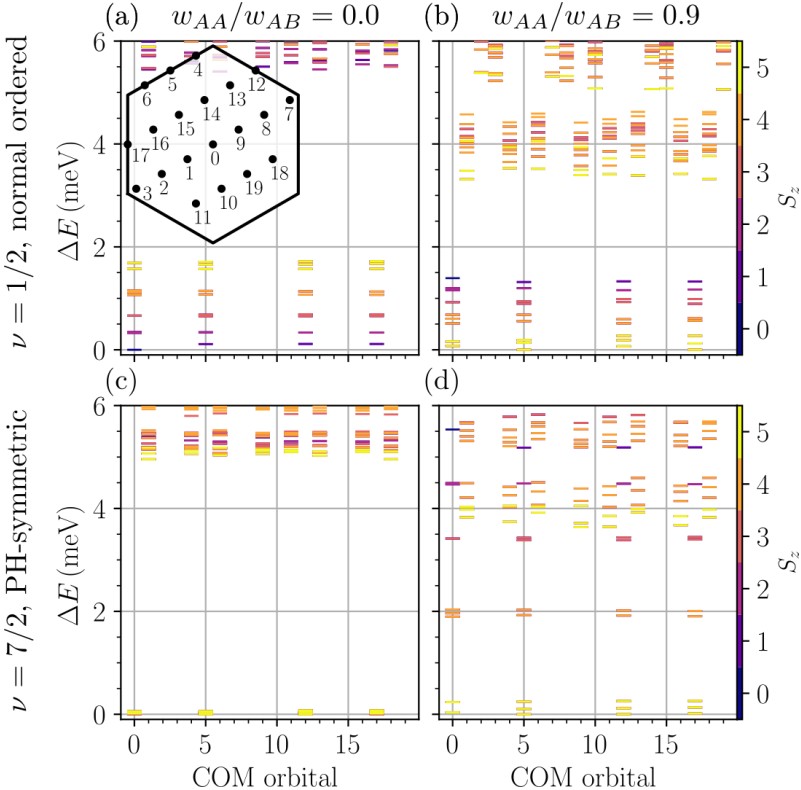

Figure 9: Spin resolved many-body spectrum of TDBG for pure interactions at $\eta = 0$ on the $N = 20$ (C20) cluster. The system is at filling $\nu = 1/2$ [$\nu = 7/2$] with normal ordered $H_{\mathrm{int}}^{\mathrm{NO}}$ [particle-hole symmetric $H_{\mathrm{int}}^{\mathrm{PH}}$] interactions at (a) [(c)] $w_{AA}/w_{AB} = 0.0$ as well as (b) [(d)] $w_{AA}/w_{AB} = 0.9$. The slight asymmetry of the energy levels at different $\mathbf{M}_j$ points is rooted in the lack of $C_3$ symmetry of the simulation cluster C20.

contrast to $C_3$ symmetric lattices C12 and C16, the lack of threefold rotational symmetry on the cluster C20 renders the absolutely lowest energy state ferromagnetic, more specifically the $\mathrm{TCDW_N}$. Tuning $\eta > 0$, hence introducing a finite dispersion, the singlet and maximally polarized levels develop a significant energetic splitting in favor of the TAF such that the ground state becomes antiferromagnetic again. The manifold of TCDWs, on the other hand, does not appear to be destabilized by switching from $H_{\mathrm{int}}^{\mathrm{NO}}$ to $H_{\mathrm{int}}^{\mathrm{PH}}$ when comparing Fig. 9 (b) with Fig. 9 (d). Quite the contrary, putting the splitting of the $2J + 1$-dimensional manifolds of ferromagnetic levels in relation to the gap to the spin multiplet with $S = J - 1$ we find the relative anisotropies to be substantially lowered compared to the case of normal ordered interactions.

The persistence of the suspected spin polarized O(3)-TCDW manifold is verified in Fig. 10 up to $N = 36$. The quasi-degeneracy of $2J + 1 = 19$ levels with momentum transfer $\mathbf{q} = \mathbf{M}_j$ is consistent among TDBG (ABAB) as well as TMBG with an original moiré Chern number $|C| = 2$, while it is lacking for ABBA stacked TDBG with $|C| = 1$. Especially the data for TMBG in Fig. 10 (c) suggests a low degree of anisotropy in the O(3) charge density wave channel, indicated by the small splitting of the ground state levels. On the other hand, Fig. 10 (d) does not fit the discussed pattern but only has a four-fold degeneracy across COM momenta $\mathbf{\Gamma}$ and $\mathbf{M}_j$, which could be associated to a four-fold extension of the unit cell via ordinary CDW order. However, we could not establish agreement between the evidence for symmetry breaking coming from the low-energy ED spectrum and the self-consistent order parameter obtained from an unrestricted HF treatment. The signatures were found to be less consistent

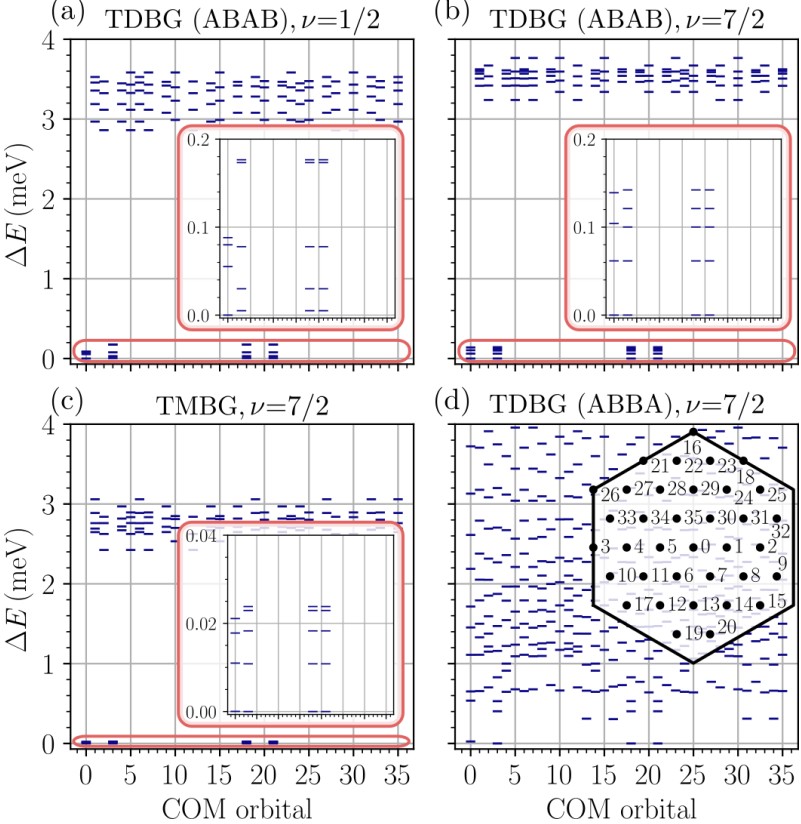

Figure 10: Completely flavor polarized many-body spectra at $w_{AA}/w_{AB} = 0.9$ for (a) $H_{\text{int}}^{\text{NO}}$ with $\nu = 1/2$ of ABAB stacked TDBG as well as $H_{\text{int}}^{\text{PH}}$ with $\nu = 7/2$ for (b) ABAB stacked TDBG, (c) TMBG and (d) ABBA stacked TDBG on cluster C36.

across the phase diagram than for the set of models studied in the frame of this paper and thus a thorough analysis is left to future works. One likely reason for this discrepancy may be the effect of the ABBA stacking chirality on the quantum geometry of the moiré Bloch vectors. Despite having almost identical band structures at $w_{AA}/w_{AB} = 0.9$ (see Fig. 12), the ABBA stacking conduction band differs drastically in its Berry curvature and consequently the Chern number of $|C| = 1$, opposed to ABAB stacked TDBG and the studied configuration of TMBG with $|C| = 2$. The stark contrast in quantum geometry may be at least co-responsible for the lack of TAF stabilization, whose tower of states may be intimately connected to the emergence of the approximately $O(3)$ symmetric manifold of ferromagnetic TCDWs.

## A.2 Hartree-Fock Berry curvatures

Since the HF approximation to the interaction Hamiltonian defines a quadratic problem in momentum space, the resulting occupied bands may be straightforwardly characterized by a composite Chern number. At a filling of $\nu = 1/2$, all electrons are accommodated in two out of 16 bands per momentum in the reduced (moiré) Brillouin zone (RBZ), which collectively contribute to the Berry curvatures displayed in Fig. 11. For reference, we also include the Berry curvature of the original moiré conduction band folded by the momentum transfers $\mathbf{q} \in \{0, \mathbf{M}_1, \mathbf{M}_2, \mathbf{M}_3\}$. Apparently, all $|C| = 1$ phases inherit a lot of their quantum geometry from the underlying $|C| = 2$ band. The relatively homogeneous distribution at $w_{AA}/w_{AB} = 0.0$ is preserved just like the very peaked nature at $w_{AA}/w_{AB} = 0.9$. At both relaxation values, the TAF exhibits the least fluctuations in $\mathcal{F}(\mathbf{k})$ while the TCDW$_N$ varies the most. Nevertheless, albeit the TAF and the ferromagnetic TCDWs are phenomenologically very distinct, the results

of Fig. 11 suggest a more intimate relationship — potentially enabled by the Chern character of the original band. This conjecture is further corroborated by analyzing the deviation $\Delta\mathcal{F}(\mathbf{k}) = \mathcal{F}^{\mathrm{HF}}(\mathbf{k}) - \frac{1}{2}\mathcal{F}^{\mathrm{ori}}(\mathbf{k})$ (not shown here), where $\mathcal{F}^{\mathrm{HF}}$ is the Berry curvature obtained via HF and $\mathcal{F}^{\mathrm{ori}}$ the one from the original moiré band. Apart from minor quantitative differences and the lowered symmetry of the $\mathrm{TCDW_N}$, the overarching feature of flux balancing from high $\mathcal{F}(\mathbf{k})$ regions to those of low Berry curvature is found to be prevalent across all identified phases.

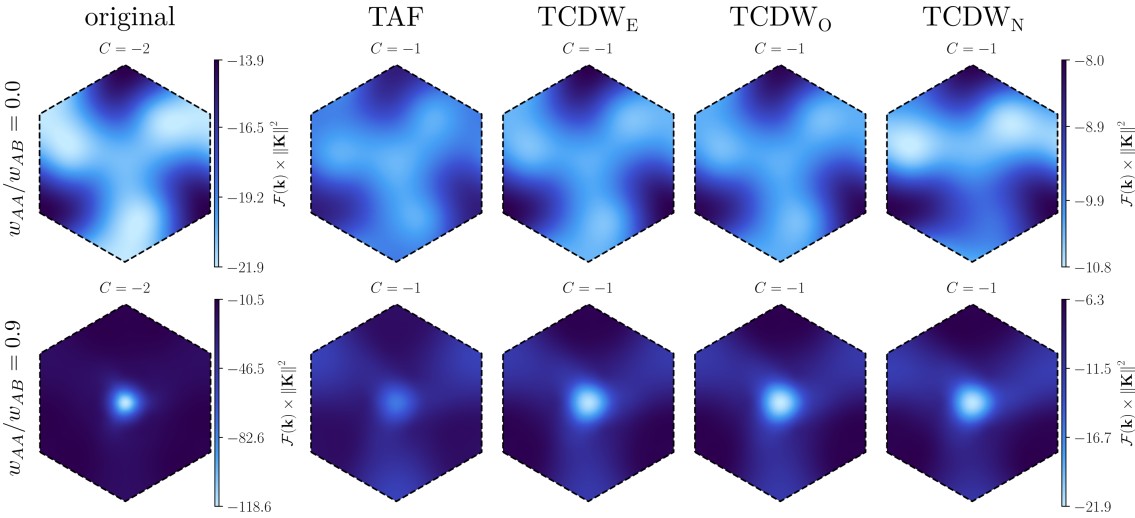

Figure 11: Folded Berry curvature to the reduced Brillouin zone of the original TDBG (ABAB) conduction band compared to those of self-consistent HF solutions at two representative values of $w_{AA}/w_{AB}$. The used cluster is C576.

## B Moiré Chern band models

On the single-particle level, we adopt the continuum formulation of the TDBG model at small twist angles $\theta$ as it is desribed in Ref. [39]. Defining $\mathbf{k}_l = R(\pm\theta/2)(\mathbf{k} - \mathbf{K}_\tau^l)$ and $k_\pm = \tau k_x \pm i k_y$, with $\mathbf{K}_\tau^l$ denoting the graphene Dirac momentum of valley $\tau = \pm$ on moiré layer $l = 1, 2$, the Hamiltonian is written in the $(A_1, B_1, A_2, B_2, A_3, B_3, A_4, B_4)$-basis as

$$H_{\mathrm{TDBG}}^{\mathrm{ABAB}} = \begin{pmatrix} H_0(\mathbf{k}_1) & g^\dagger(\mathbf{k}_1) & & \\ g(\mathbf{k}_1) & H_0'(\mathbf{k}_1) & U^\dagger & \\ & U & H_0(\mathbf{k}_2) & g^\dagger(\mathbf{k}_2) \\ & & g(\mathbf{k}_2) & H_0'(\mathbf{k}_2) \end{pmatrix} + V. \tag{11}$$

The terms appearing in $H_{\mathrm{TDBG}}^{\mathrm{ABAB}}$ are

$$H_0(\mathbf{k}) = \begin{pmatrix} 0 & -\hbar v_0 k_- \\ -\hbar v_0 k_+ & \Delta' \end{pmatrix}, \quad H_0'(\mathbf{k}) = \begin{pmatrix} \Delta' & -\hbar v_0 k_- \\ -\hbar v_0 k_+ & 0 \end{pmatrix}, \quad g(\mathbf{k}) = \begin{pmatrix} \hbar v_4 k_+ & t_1 \\ \hbar v_3 k_- & \hbar v_4 k_+ \end{pmatrix}, \tag{12}$$

corresponding to contributions from the mono- and bilayer graphene Hamiltonians with parameters $(t_0, t_1, t_3, t_4, \Delta') = (2610, 361, 283, 138, 15)\,\mathrm{meV}$ [63], giving $\hbar v_i = 3/2 t_i a_0$ $(i = 0, 3, 4)$ with $a_0$ denoting the graphene nearest-neighbor distance. A finite displacement field $V = \mathrm{diag}(\frac{1}{2}\Delta\mathbb{1}, \frac{1}{6}\Delta\mathbb{1}, -\frac{1}{6}\Delta\mathbb{1}, -\frac{1}{2}\Delta\mathbb{1})$ breaks the point group $D_3$ of the lattice down to $C_3$

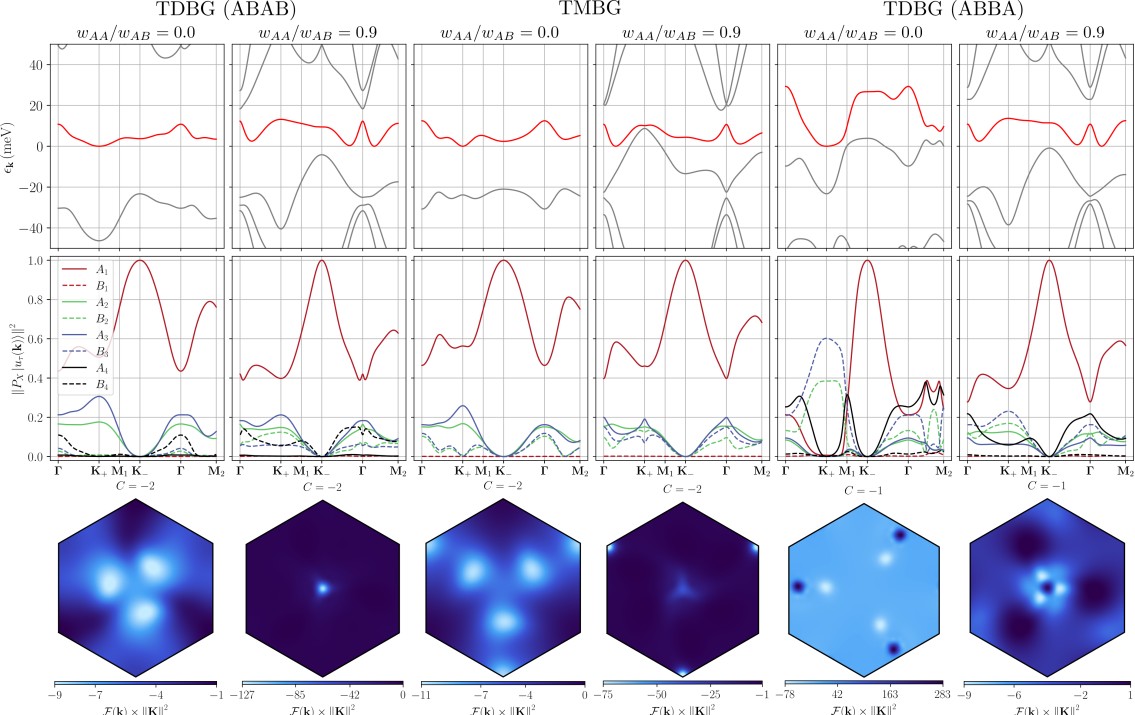

Figure 12: Band structure near the Fermi level (top), sublattice polarization (middle) and Berry curvature (bottom) of the $\tau = -$ conduction band of ABAB stacked TDBG, TMBG as well as ABBA stacked TDBG at two representative values for $w_{AA}/w_{AB}$.

and can be used to energetically isolate the conduction band in each valley. We set $\Delta = 50\,\text{meV}$ throughout our work. The moiré interlayer hopping matrix $U$ is given by

$$U = \begin{pmatrix} w_{AA} & w_{AB} \\ w_{AB} & w_{AA} \end{pmatrix} + \begin{pmatrix} w_{AA} & w_{AB}\omega^{-\tau} \\ w_{AB}\omega^{\tau} & w_{AA} \end{pmatrix} e^{i\tau \mathbf{G}_1^M \cdot \mathbf{r}} + \begin{pmatrix} w_{AA} & w_{AB}\omega^{\tau} \\ w_{AB}\omega^{-\tau} & w_{AA} \end{pmatrix} e^{i\tau(\mathbf{G}_1^M + \mathbf{G}_2^M) \cdot \mathbf{r}}, \quad (13)$$

where $\omega = e^{i2\pi/3}$ and $\mathbf{G}_i^M$ are the moiré reciprocal lattice vectors. The inter sublattice hopping amplitude $w_{AB}$ is taken to be $w_{AB} = 100\,\text{meV}$, whereas the relaxation ratio $w_{AA}/w_{AB}$ is used as a tuning parameter in our studies. Our numerical diagonalization of Eq. (11) is performed in momentum space over a finite number of wave vectors within a cutoff circle centered around the midpoint of $\mathbf{K}_\tau^1$ and $\mathbf{K}_\tau^2$. For band structure and Berry curvature computations the cutoff radius $r_c$ is chosen to be $r_c = 4\|\mathbf{G}^M\|$, while in the computation of matrix elements for HF we used $r_c = 6\|\mathbf{G}^M\|$ and for ED $r_c = 8\|\mathbf{G}^M\|$ to ensure that Hermiticity is respected sufficiently accurately in our numerics. As mentioned in Ref. [39], the TDBG Hamiltonian for the ABBA stacking variant may be obtained from Eq. (11) by swapping $H_0(\mathbf{k}_2)$ with $H'_0(\mathbf{k}_2)$ as well as $g(\mathbf{k}_2)$ and $g^\dagger(\mathbf{k}_2)$. Furthermore, the TMBG Hamiltonian is contained in Eq. (11) by restricting to a $6 \times 6$ sub-matrix of components $(A_1, B_1, A_2, B_2, A_3, B_3)$. The TDBG systems are studied at $\theta = 1.3°$ while for TMBG we set $\theta = 1.2°$.

As can be seen from the band structures displayed in the top row of Fig. 12, the chosen parameter values result in an isolated conduction band where, among other quantities, the degree of separation from other bands depends on $w_{AA}/w_{AB}$. With the exception of ABBA stacked TDBG, an analysis of the conduction band Bloch vectors in Fig. 12 (panels in middle row) yielded a substantial degree of sublattice polarization for the studied systems, which is in accordance with previous works [10, 64].

In our measurements of observables we thus project the form factors

$\Lambda^{\tau}(\mathbf{k}, \mathbf{q})$ to the topmost layer, more specifically the dominant sublattice $A_1$, giving $\lambda^{\tau}(\mathbf{k}, \mathbf{q}) = \langle P_{A_1} u_{\tau}(\mathbf{k}) | P_{A_1} u_{\tau}(\mathbf{k} + \mathbf{q}) \rangle$. Without the projection to the top layer, certain signatures in the measured observables were found to be blurred.

We numerically computed the Berry curvature $\mathcal{F}(\mathbf{k})$ across the moiré Brillouin zone and subsequently the Chern number using the technique devised in Ref. [65]. We quantify the homogeneity of $\mathcal{F}(\mathbf{k})$ using its root-mean-square error

$$\text{RMSE}(\mathcal{F}(\mathbf{k}) \times \|\mathbf{K}\|^2) = \sqrt{\frac{1}{N} \sum_{i=1}^{N} \mathcal{F}(\mathbf{k}_i) \times \|\mathbf{K}\|^2}, \tag{14}$$

where $\mathbf{k}_i$ are the momenta of the discretized MBZ and the squared magnitude of the moiré Dirac momentum $\|\mathbf{K}\|^2$ is used to obtain a dimensionless quantity. As is evident in Fig. 12, the $|C| = 2$ bands of ABAB stacked TDBG and TMBG exhibit a relatively homogeneous $\mathcal{F}(\mathbf{k})$ at $w_{AA}/w_{AB} = 0$, which gradually becomes more peaked towards $w_{AA}/w_{AB} = 0.9$. This is in stark contrast to the behavior of the $|C| = 1$ band of ABBA stacked TDBG, whose Berry curvature is highly inhomogeneous at $w_{AA}/w_{AB} = 0$ but smoothens out towards higher $w_{AA}/w_{AB}$. The remarkable property of almost identical band structures near $w_{AA}/w_{AB} = 0.9$, yet apparently so different quantum geometries was already mentioned in previous works [18,39]. It should be noted, that an inversion of the displacement field $\Delta \to -\Delta$ has a profound impact on the TMBG model [40]. The conduction band no longer has $|C| = 2$, but $|C| = 1$ instead, accompanied by a lack of sublattice polarization. The inequivalence of the $\Delta > 0$ and $\Delta < 0$ regimes has also been observed in experiments [7,10]. Since the experimental signatures relevant to our discussion appear in the $|C| = 2$ regime [19] and the altered properties of the Bloch wave functions pose a rather different situation, we leave the discussion of the $|C| = 1$ model to future works.

## C Methods

### C.1 Exact diagonalization

Our primary method of choice for the analysis of Eq. (1) is exact diagonalization in momentum space on finite systems with periodic boundary conditions, which has previously been applied to study other graphene moiré systems [45, 46, 52, 66–69]. ED provides us with an unbiased, numerically exact way of obtaining the many-body wave functions and full set of low-energy levels, which collectively constitute a spectral fingerprint for the prevalent ordered phases. In combination with group theoretical tools, this constitutes a powerful machinery for the identification of symmetry broken, but also topological phases [24, 42–44, 70] for various system geometries. Although well established in ED studies of spin Hamiltonians, the analysis of spontaneously broken continuous symmetries via the tower of states method isn't commonly performed in the context of fractionally filled bands. The challenge of exponentially increasing Hilbert spaces is mitigated by decomposing them into separate symmetry sectors, projecting to the most relevant electronic bands and exploiting the polarization tendencies of the studied systems. For the diagonalization we make use of our own implementation of the Lanczos algorithm as well as an ARPACK [71] based Arnoldi iteration to obtain the spectrum and wave functions of the lowest energy levels. For Hilbert space dimensions up to $\sim 10^6$, we converge the ten lowest eigenvalues via the ARPACK implementation, otherwise we use ordinary Lanczos. Working in the spin and valley resolved Hilbert space of a single band, the Hamiltonian of Eq. (1) is composed of the diagonal kinetic part

$$H_{\text{kin}} = \sum_{\sigma=\uparrow\downarrow} \sum_{\tau=\pm} \sum_{\mathbf{k}\in\text{MBZ}} \epsilon_{\tau,\mathbf{k}} c^\dagger_{\sigma,\tau,\mathbf{k}} c_{\sigma,\tau,\mathbf{k}} \tag{15}$$

as well as the normal ordered band-projected density-density interaction

$$
\begin{aligned}
H_{\text{int}}^{\text{NO}} =& \frac{1}{2\Omega} \sum_{\mathbf{q}\in\mathbb{R}^2} V(\mathbf{q}) : \rho_{-\mathbf{q}}\rho_{\mathbf{q}} : \\
=& \frac{1}{2\Omega} \sum_{\substack{\sigma,\sigma' \\ \tau,\tau'}} \sum_{\substack{\mathbf{q}\in\mathbb{R}^2 \\ \mathbf{k},\mathbf{k}'\in\text{MBZ}}} V(\mathbf{q})\Lambda^\tau(\mathbf{k},-\mathbf{q})\Lambda^{\tau'}(\mathbf{k}',\mathbf{q}) c^\dagger_{\sigma,\tau,\mathbf{k}} c^\dagger_{\sigma',\tau',\mathbf{k}'} c_{\sigma',\tau',\mathbf{k}'+\mathbf{q}} c_{\sigma,\tau,\mathbf{k}-\mathbf{q}} \\
=& \frac{1}{2\Omega} \sum_{\substack{\sigma,\sigma' \\ \tau,\tau'}} \sum_{\substack{\mathbf{q},\mathbf{k}, \\ \mathbf{k}'\in\text{MBZ}}} \underbrace{\sum_{\mathbf{G}\in\text{MRL}} V(\mathbf{q}+\mathbf{G})\Lambda^\tau(\mathbf{k},-\mathbf{q}-\mathbf{G})\Lambda^{\tau'}(\mathbf{k}',\mathbf{q}+\mathbf{G})}_{V^{\tau,\tau'}(\mathbf{k},\mathbf{k}',\mathbf{q})} c^\dagger_{\sigma,\tau,\mathbf{k}} c^\dagger_{\sigma',\tau',\mathbf{k}'} c_{\sigma',\tau',\mathbf{k}'+\mathbf{q}} c_{\sigma,\tau,\mathbf{k}-\mathbf{q}} \\
=& \frac{1}{2\Omega} \sum_{\sigma,\sigma',\tau,\tau'} \sum_{\mathbf{q},\mathbf{k},\mathbf{k}'} V^{\tau,\tau'}(\mathbf{k},\mathbf{k}',\mathbf{q}) c^\dagger_{\sigma,\tau,\mathbf{k}} c^\dagger_{\sigma',\tau',\mathbf{k}'} c_{\sigma',\tau',\mathbf{k}'+\mathbf{q}} c_{\sigma,\tau,\mathbf{k}-\mathbf{q}},
\end{aligned}
\tag{16}
$$

with MRL denoting the moiré reciprocal lattice. The form factors $\Lambda^\tau(\mathbf{k},\mathbf{q}) = \langle u_\tau(\mathbf{k})|u_\tau(\mathbf{k}+\mathbf{q})\rangle$ are defined by overlaps of the conduction band Bloch functions $|u_\tau(\mathbf{k})\rangle$ and the Yukawa-type Coulomb potential $V(\mathbf{q}) = (e^2/2\epsilon\epsilon_0\sqrt{\|\mathbf{q}\|^2+\kappa^2})$ is used with $\epsilon = 4$ and a screening length equal to the moiré period $1/\kappa = L^{\text{M}}$.

Introducing the modified density operators $\delta\rho_{\mathbf{q}} = \rho_{\mathbf{q}} - \frac{1}{2}\sum_{\mathbf{k}}\text{Tr}[\Lambda(\mathbf{k},\mathbf{q})]\delta_{\mathbf{q}\in\text{MRL}}$ the interaction becomes

$$
\begin{aligned}
H_{\text{int}}^{\text{PH}} =& \frac{1}{2\Omega} \sum_{\mathbf{q}\in\mathbb{R}^2} V(\mathbf{q})\delta\rho_{-\mathbf{q}}\delta\rho_{\mathbf{q}} \\
=& \frac{1}{2\Omega} \sum_{\substack{\sigma,\sigma' \\ \tau,\tau'}} \sum_{\substack{\mathbf{q},\mathbf{k} \\ \mathbf{k}'\in\text{MBZ}}} V^{\tau,\tau'}(\mathbf{k},\mathbf{k}',\mathbf{q}) \left[ c^\dagger_{\sigma,\tau,\mathbf{k}} c_{\sigma,\tau,\mathbf{k}-\mathbf{q}} - \frac{1}{2}\delta_{\mathbf{q},0} \right] \left[ c^\dagger_{\sigma',\tau',\mathbf{k}'} c_{\sigma',\tau',\mathbf{k}'+\mathbf{q}} - \frac{1}{2}\delta_{\mathbf{q},0} \right] \\
=& \frac{1}{2\Omega} \sum_{\sigma,\sigma',\tau,\tau'} \sum_{\mathbf{q},\mathbf{k},\mathbf{k}'} V^{\tau,\tau'}(\mathbf{k},\mathbf{k}',\mathbf{q}) \bigg[ c^\dagger_{\sigma,\tau,\mathbf{k}} c_{\sigma,\tau,\mathbf{k}-\mathbf{q}} c^\dagger_{\sigma',\tau',\mathbf{k}'} c_{\sigma',\tau',\mathbf{k}'+\mathbf{q}} \\
& \qquad\qquad\qquad - \frac{1}{2}(c^\dagger_{\sigma,\tau,\mathbf{k}} c_{\sigma,\tau,\mathbf{k}-\mathbf{q}} + c^\dagger_{\sigma',\tau',\mathbf{k}'} c_{\sigma',\tau',\mathbf{k}'+\mathbf{q}})\delta_{\mathbf{q},0} + \frac{1}{4}\delta_{\mathbf{q},0} \bigg] \\
=& \frac{1}{2\Omega} \sum_{\sigma,\sigma',\tau,\tau'} \sum_{\mathbf{q},\mathbf{k},\mathbf{k}'} V^{\tau,\tau'}(\mathbf{k},\mathbf{k}',\mathbf{q}) c^\dagger_{\sigma,\tau,\mathbf{k}} c^\dagger_{\sigma',\tau',\mathbf{k}'} c_{\sigma',\tau',\mathbf{k}'+\mathbf{q}} c_{\sigma,\tau,\mathbf{k}-\mathbf{q}} \\
& \qquad + \sum_{\sigma,\tau,\mathbf{k}} \frac{1}{2\Omega} \left[ \sum_{\mathbf{k}'} V^{\tau,\tau}(\mathbf{k},\mathbf{k}',\mathbf{k}-\mathbf{k}') - 2\sum_{\tau'} V^{\tau,\tau'}(\mathbf{k},\mathbf{k}',0) \right] c^\dagger_{\sigma,\tau,\mathbf{k}} c_{\sigma,\tau,\mathbf{k}} + \text{const.} \\
=& H_{\text{int}}^{\text{NO}} + \frac{1}{2} \sum_{\sigma,\tau,\mathbf{k}} E_h^\tau(\mathbf{k}) c^\dagger_{\sigma,\tau,\mathbf{k}} c_{\sigma,\tau,\mathbf{k}} + \text{const.}.
\end{aligned}
\tag{17}
$$

The constant offset in Eq. (17) is neglected in our calculations and $E_h^\tau(\mathbf{k})$ constitutes an additional, interaction induced dispersion whose bandwidth typically exceeds the one of the moiré flat bands. It compensates for the very same term appearing after the particle-hole transformation $c \to h^\dagger$, such that at $\eta = 0$ electrons and holes are (up to a physically irrelevant energetic shift) subject to the same Hamiltonian.

Despite the projection to a single band, a complete ED treatment of all orbital and flavor degrees of freedom is only feasible for very small lattices. In order to extend our results to larger system sizes, we exploit the flavor polarization tendencies assessed on smaller clusters (and supported by HF) by restricting the dynamics to a subset of spin and valley degrees of freedom. On the electron side ($\nu = 1/2$), when all but the active polarized orbitals are empty, we may simply neglect the terms with different flavor without changing the physics at play. On the other hand, at $\nu = 7/2$, when all except for a $\nu_h = 1/2$ subset of Fock orbitals are occupied, this procedure misses an additional dispersive term stemming from the normal ordered, quartic creation and annihilation operators acting on the vacuum for holes instead of the one for electrons. Without loss of generality, let $\tau_d = -$ be the valley quantum number of dynamic orbitals while all the ones in $\tau_f = +$ are fully occupied and thus inactive. Acting with $H_{\mathrm{int}}^{\mathrm{NO}}$ on a state from this symmetry sector $|\psi\rangle_- = |11\cdots1\rangle_+ |\phi\rangle_-$ gives

$$
\begin{aligned}
H_{\mathrm{int}}^{\mathrm{NO}}|\psi\rangle_- =& \frac{1}{2\Omega}\sum_{\sigma,\sigma',\tau,\tau'}\sum_{\mathbf{q},\mathbf{k},\mathbf{k}'} V^{\tau,\tau'}(\mathbf{k},\mathbf{k}',\mathbf{q})c^\dagger_{\sigma,\tau,\mathbf{k}}c^\dagger_{\sigma',\tau',\mathbf{k}'}c_{\sigma',\tau',\mathbf{k}'+\mathbf{q}}c_{\sigma,\tau,\mathbf{k}-\mathbf{q}}|\psi\rangle_- \\
=& \frac{1}{2\Omega}\sum_{\sigma,\sigma'}\sum_{\mathbf{q},\mathbf{k},\mathbf{k}'}|11\cdots1\rangle_+ \Big[\delta_{\mathbf{q},\mathbf{0}}V^{+,-}(\mathbf{k},\mathbf{k}',\mathbf{0})c^\dagger_{\sigma',-,\mathbf{k}'}c_{\sigma',-,\mathbf{k}'} + \delta_{\mathbf{q},\mathbf{0}}V^{-,+}(\mathbf{k},\mathbf{k}',\mathbf{0})c^\dagger_{\sigma,-,\mathbf{k}}c_{\sigma,-,\mathbf{k}} \\
& + V^{-,-}(\mathbf{k},\mathbf{k}',\mathbf{q})c^\dagger_{\sigma,-,\mathbf{k}}c^\dagger_{\sigma',-,\mathbf{k}'}c_{\sigma',-,\mathbf{k}'+\mathbf{q}}c_{\sigma,-,\mathbf{k}-\mathbf{q}} + \mathrm{const.}\Big]|\phi\rangle_- \\
=& |11\cdots1\rangle_+ \Big[\frac{1}{2\Omega}\sum_{\sigma,\sigma'}\sum_{\mathbf{q},\mathbf{k},\mathbf{k}'}V^{-,-}(\mathbf{k},\mathbf{k}',\mathbf{q})c^\dagger_{\sigma,-,\mathbf{k}}c^\dagger_{\sigma',-,\mathbf{k}'}c_{\sigma',-,\mathbf{k}'+\mathbf{q}}c_{\sigma,-,\mathbf{k}-\mathbf{q}} \\
& + \frac{2}{\Omega}\sum_{\sigma,\mathbf{k}}\sum_{\mathbf{k}'}V^{-,+}(\mathbf{k},\mathbf{k}',\mathbf{0})c^\dagger_{\sigma,-,\mathbf{k}}c_{\sigma,-,\mathbf{k}} + \mathrm{const.}\Big]|\phi\rangle_- \\
=& |11\cdots1\rangle_+ \Big[H_{\mathrm{int}}^{\mathrm{NO},-} + H_{d,f}^{-,+} + \mathrm{const.}\Big]|\phi\rangle_- \,.
\end{aligned}
\tag{18}
$$

If spin polarization is also present, say to $S_z$ sector $\sigma = \downarrow$, Eq. (18) becomes

$$
\begin{aligned}
H_{\mathrm{int}}^{\mathrm{NO}}|\psi\rangle_{\downarrow,-} =& |11\cdots1\rangle_{+,(\uparrow,-)}\Big[\frac{1}{2\Omega}\sum_{\mathbf{q},\mathbf{k},\mathbf{k}'}V^{-,-}(\mathbf{k},\mathbf{k}',\mathbf{q})c^\dagger_{\downarrow,-,\mathbf{k}}c^\dagger_{\downarrow,-,\mathbf{k}'}c_{\downarrow,-,\mathbf{k}'+\mathbf{q}}c_{\downarrow,-,\mathbf{k}-\mathbf{q}} \\
& + \frac{1}{\Omega}\sum_{\mathbf{k}}\sum_{\mathbf{k}'}\big[2V^{-,+}(\mathbf{k},\mathbf{k}',\mathbf{0}) + V^{-,-}(\mathbf{k},\mathbf{k}',\mathbf{0})\big]c^\dagger_{\downarrow,-,\mathbf{k}}c_{\downarrow,-,\mathbf{k}} + \mathrm{const.}\Big]|\phi\rangle_{\downarrow,-} \\
=& |11\cdots1\rangle_{+,(\uparrow,-)}\Big[H_{\mathrm{int}}^{\mathrm{NO},\downarrow,-} + H_{d,f}^{(\downarrow,-),[+,(\uparrow,-)]} + \mathrm{const.}\Big]|\phi\rangle_{\downarrow,-}\,.
\end{aligned}
\tag{19}
$$

The incorporated flavor degrees of freedom per cluster and the resulting Hilbert space dimensions are listed in Tab. 1.

For a given number of electrons $N_e$, determining the band filling as $\nu = N_e/N$, we calculate the chemical potential as

$$
\mu = \min_{\delta N_e > 0}\frac{E_0(N_e + \delta N_e) - E_0(N_e)}{\delta N_e}\,.
\tag{20}
$$

Plotting the band filling $\nu$ over the chemical potential $\mu$ may be used to infer the conductive properties of the prevalent phase. Large steps in $\mu$ while $\nu$ remains constant indicate electronic gaps which render the system insulating, while a compressible, metallic state is characterized by a rather smooth evolution of $\nu$ with $\mu$.

The generic structure factor on the top layer is defined as

Table 1: Overview of simulation cells used for ED in this work. All clusters were chosen to have three **M** points to support all observed forms of symmetry breaking. Depending on the number of momentum sites $N$, center of mass momentum $\mathbf{k}_{\text{COM}}$, spin $S_z$ and valley polarization $N_v$ resolved calculations could be performed. The total and maximum symmetry-decomposed, diagonalized Hilbert space dimensions at $\nu = 1/2$ (7/2) are given for the respectively incorporated degrees of freedom.

| ID | $N$ | Torus $[[a,b],[c,d]]$ | Aspect ratio | Point group | $\mathbf{k}_{\text{COM}}$ | $S_z$ | $N_v$ | HS dim. total | HS dim. diag. |
|---|---|---|---|---|---|---|---|---|---|
| C12 | 12 | $[[2,2],[2,-4]]$ | 1.0 | $D_6$ | ✓ | ✓ | ✓ | $1.23{\times}10^7$ | $5.23{\times}10^4$ |
| C16 | 16 | $[[4,0],[0,4]]$ | 1.0 | $D_6$ | ✓ | ✓ | ✗ | $1.05{\times}10^7$ | $2.07{\times}10^5$ |
| C20 | 20 | $[[2,2],[4,-6]]$ | 1.5 | $D_2$ | ✓ | ✓ | ✗ | $8.48{\times}10^8$ | $1.20{\times}10^7$ |
| C28 | 28 | $[[2,4],[6,-2]]$ | 1.0 | $C_6$ | ✓ | ✗ | ✗ | $4.01{\times}10^7$ | $1.43{\times}10^6$ |
| C36 | 36 | $[[6,0],[0,6]]$ | 1.0 | $D_6$ | ✓ | ✗ | ✗ | $9.08{\times}10^9$ | $2.52{\times}10^8$ |

$$S^\alpha(\mathbf{q}) = \frac{1}{N}\langle \tilde{\rho}_{\alpha,-\mathbf{q}}\tilde{\rho}_{\alpha,\mathbf{q}}\rangle_{\Psi_0} = \frac{1}{N}\left\|\tilde{\rho}_{\alpha,\mathbf{q}}\,|\Psi_0\rangle\right\|^2 = \frac{1}{N}\left\|\sum_\tau\sum_{\mathbf{k}}\lambda^\tau(\mathbf{k},\mathbf{q})\mathbf{c}^\dagger_{\tau,\mathbf{k}}\sigma_\alpha\mathbf{c}_{\tau,\mathbf{k}+q}\,|\Psi_0\rangle\right\|^2, \quad (21)$$

with $S_C(\mathbf{q})\equiv S^0(\mathbf{q})$ sensitive to charge- and $S_S(\mathbf{q})\equiv S^z(\mathbf{q})$ to spin order. Since $S_C(\mathbf{0})$ simply sums the charge density, it is neglected in our figures. Although the definition of the structure factor in Eq. (21) is valid beyond the first moiré Brillouin zone, corresponding to sub-moiré scale order, the dominant peaks were always found to be located inside the MBZ (including its border).

Since the full spectrum of the many-body Hamiltonian forms an orthonormal basis of the Hilbert space, one may generally decompose an arbitrary state $|\Phi\rangle$ over this set of eigenvectors. However, similar to the computation of the spectrum itself, the large dimension of the Hilbert space renders a decomposition over the complete eigenspace infeasible. Instead, akin to the Lanczos method, we make use of the iterative construction of Krylov subspaces to obtain the most dominant amplitudes [72]. Given $|\Phi\rangle$, it is first projected to a certain symmetry sector $\alpha$ and subsequently the resulting state $|P_\alpha\Phi\rangle$ is used as a start vector for the construction of the Krylov space $\{H^n|P_\alpha\Phi\rangle\}_n$. A diagonalization of the associated T-matrix yields the weights of the decomposition $I_{\alpha,n} = \left|\langle P_\alpha\Phi|\psi_{\alpha,n}\rangle\right|^2$ in the basis of the many-body levels $\left|\psi_{\alpha,n}\right\rangle$ with energy $E_{\alpha,n}$. In our implementation, we take the variation of the maximum weight as a convergence criterion and contributions of levels separated by less than $10^{-7}$ eV are merged. The sharper the decomposition of the start state, i.e. the fewer levels actually contribute, the quicker the algorithm converges. This approach to the dynamic decomposition of a trial state was pioneered in Ref. [73].

The simulation clusters considered in our work are compiled in Tab. 1, along with some of their properties and ED implementation details, labeled by a unique ID of the form C$N$, where $N$ is the system size.. The square lattices marked by C144 as well as C576, with $a = d = 12/24$ and $b = c = 0$ respectively, are not used in ED, but only in HF and both feature the point group $D_6$. The torus spans the real-space simulation cell like $\mathbf{T}_1 = a\mathbf{L}_1^M + b\mathbf{L}_2^M$ and $\mathbf{T}_2 = c\mathbf{L}_1^M + d\mathbf{L}_2^M$, where $\mathbf{L}_i^M$ are the moiré lattice vectors. The momentum-space discretization may then be derived as usual by finding the respective reciprocal vectors. The incorporation of diverse simulation clusters is provided by the QuantiPy package [74].

## C.2 Hartree-Fock

Inspired by the patterns of symmetry breaking manifest in the ED spectrum we construct an effective single-particle Hamiltonian, where the original interactions enter as direct (Hartree) and exchange (Fock) contributions only and the exponential scaling of ED may be avoided at the cost of restricting the ground state to be a Slater determinant. Guided by the ED results, we allow for unrestricted order parameters of the form

$$P_{\sigma,\sigma'}^{\tau,\tau'}(\mathbf{k},\mathbf{q}) = \langle c_{\sigma,\tau,\mathbf{k}}^{\dagger} c_{\sigma',\tau',\mathbf{k}+\mathbf{q}} \rangle, \tag{22}$$

with $\mathbf{q} \in \{\mathbf{0}, \mathbf{M}_1, \mathbf{M}_2, \mathbf{M}_3\}$. The mean-field approximation to the quartic interaction Hamiltonian of Eq. (16) gives

$$H_{\text{int}}^{\text{NO,HF}} = \sum_{\mathbf{k}\in\text{MBZ}} \sum_{\mathbf{q}} \mathbf{c}_{\mathbf{k}}^{\dagger} F^{[P]}(\mathbf{k},\mathbf{q}) \mathbf{c}_{\mathbf{k}+\mathbf{q}} - \frac{1}{2}\operatorname{Tr}\left(F^{[P]}P\right), \tag{23}$$

with

$$\begin{aligned}
F_{\sigma,\sigma',\tau,\tau'}^{[P]}(\mathbf{k},\mathbf{q}) = &\frac{1}{\Omega}\sum_{\mathbf{k}'}\delta_{\sigma,\sigma'}\delta_{\tau,\tau'}\sum_{\tilde{\sigma},\tilde{\tau}}V^{\tau,\tilde{\tau}}(\mathbf{k},\mathbf{k}',-\mathbf{q})P_{\tilde{\sigma},\tilde{\sigma}}^{\tilde{\tau},\tilde{\tau}}(\mathbf{k}',-\mathbf{q}) \\
&- V^{\tau,\tau'}(\mathbf{k},\mathbf{k}',\mathbf{k}-\mathbf{k}'+\mathbf{q})P_{\sigma',\sigma}^{\tau',\tau}(\mathbf{k}',-\mathbf{q}),
\end{aligned} \tag{24}$$

where $F^{[P]}$ is the Fock matrix and $\frac{1}{2}\operatorname{Tr}\left(F^{[P]}P\right)$ is the energy of the condensate. The complete, tunable HF form of the normal ordered Hamiltonian is then $H^{\text{HF}} = \eta H_{\text{kin}} + (1-\eta)H_{\text{int}}^{\text{NO,HF}}$ and the corresponding particle-hole symmetric version is obtained by simply adding the additional quadratic term. Since the momenta $\mathbf{M}_j$ along with $\mathbf{\Gamma}$ form a closed group under addition modulo reciprocal lattice vectors $\mathbf{G}$, the Hamiltonian in Eq. (23) is block diagonal as a $16\times16$ matrix over a reduced Brillouin zone containing $N/4$ momenta. In order to find a self-consistent solution to the HF Hamiltonian, we start from a random density matrix and, unless mentioned otherwise, successively apply the ODA scheme described in Ref. [75].

Besides our completely unrestricted calculations of the correlation matrix $P$, to be able to select a certain subspace of solutions to iterate in we apply a symmetrization procedure. The non-symmetry-breaking state is obtained by setting all offdiagonal elements to zero, whilst equalizing spin and valley occupations in a $C_3$ symmetric way. For the TAF, $P$ is completely polarized to valley $\tau = -$ and all $\mathbf{q} = \mathbf{0}$ $\sigma_{x,y,z}$ and $\mathbf{q} \neq \mathbf{0}$ $\sigma_0$ components are made to vanish. The superposition of orthogonal spin density waves is constructed with equal magnitudes in $\sigma_x$ along $\mathbf{M}_1$, $\sigma_y$ along $\mathbf{M}_2$ and $\sigma_z$ along $\mathbf{M}_3$. Finally, the TCDW$_{\text{E,N,O}}$ is obtained by first polarizing spin and valley with maximum $\mathbf{q} = \mathbf{0}$ $\sigma_z$ and $\tau_z$ components. Subsequently the $\sigma_0$ magnitudes along all $\mathbf{q} = \mathbf{M}_j$ are either equalized and oriented in an even or odd way by previously measuring $\phi_j$ to obtain the $C_3$ symmetric TCDW$_{\text{E,O}}$ phase or a single orientation can be selected for the TCDW$_{\text{N}}$. The symmetrization is performed every second iteration after the assembly of the density matrix from the HF eigenvectors in order to judge the stationarity condition in a faithful way. Interestingly, when the chosen symmetrization does not include the solution of the truly unrestricted calculation, in some cases, the otherwise applied ODA method was found to complicate the convergence inside this subspace. For this purpose we apply the simpler Roothan algorithm (also described in Ref. [75]), which yielded more consistent results for this special scenario. Also, the use of a true step function, corresponding to the Fermi-Dirac distribution at $T = 0$, was found to be unreliable for converging the metallic non-symmetry-breaking state and we instead used the Fermi-Dirac distribution with a very small finite temperature. For all other phases the true step function was applied.

We use the following criteria to classify the ground state phases in the phase diagrams:

- TAF: $p_v = 1$, $m = 0$, $\chi > 0$,

- TCDW$_{\mathrm{E}}$: $p_v = 1$, $m = 1$, $\phi_1 \phi_2 \phi_3 > 0$, $\zeta < 0.01$,

- TCDW$_{\mathrm{O}}$: $p_v = 1$, $m = 1$, $\phi_1 \phi_2 \phi_3 < 0$, $\zeta < 0.01$,

- TCDW$_{\mathrm{N}}$: $p_v = 1$, $m = 1$, $\phi_1^2 + \phi_2^2 + \phi_3^2 > 0$, $\zeta \geqslant 0.01$,

where the valley polarization $p_v$, magnetization $m$, momentum-space scalar chirality $\chi$ as well as the CDW amplitudes $\phi_i$ are defined throughout Sec. 3 and $\zeta = \left| 1 - \sqrt[3]{\prod_j |\phi_j|} / \sqrt{\sum_j \phi_j^2 / 3} \right|$ measures the nematicity of the CDW order parameter (at $\nu = 7/2$, $p_v$ as well as $m$ are normalized by the number of holes instead of $N_e$).

In order to compute the Berry curvature and consequently the Chern number on a discretized grid, we apply the gauge invariant method described in Ref. [65]. In contrast to the computation for the original moiré flat band, due to the folding of the MBZ by $\mathbf{q} = \mathbf{M}_j$ the integration is now performed over a reduced Brillouin zone with $N/4$ sites and the additional folding index $\mathbf{Q} \in \{\mathbf{0}, \mathbf{M}_1, \mathbf{M}_2, \mathbf{M}_3\}$. Although the discussed phases are fully gapped insulators with a number of filled bands $n_F$ per momentum in the RBZ, the filled bands may be degenerate. This is e.g. the case for the TAF phase, which, due to a special spin rotation symmetry [25], has an exactly doubly degenerate HF band (for e.g. $n_F = 2$) whose Chern numbers cannot be computed individually. Instead, we turn to the composite formulation of the U(1) link variable $U_\delta(\mathbf{k})$ using

$$
\begin{aligned}
\mathcal{A}_{\boldsymbol{\delta}}^{n,n'}(\mathbf{k}) &= \sum_{\sigma,\tau,\mathbf{Q}} \Lambda^\tau(\mathbf{k} + \mathbf{Q}, \boldsymbol{\delta}) u_{n;\sigma,\tau,\mathbf{Q}}^{\mathrm{HF}*}(\mathbf{k}) u_{n';\sigma,\tau,\mathbf{Q}}^{\mathrm{HF}}(\mathbf{k} + \boldsymbol{\delta}), \\
U_{\boldsymbol{\delta}}(\mathbf{k}) &= \frac{1}{\mathcal{N}_{\boldsymbol{\delta}}(\mathbf{k})} \det\left( \mathcal{A}_{\boldsymbol{\delta}}^{n,n'}(\mathbf{k}) \right)_{n,n'=1}^{n_F},
\end{aligned}
\tag{25}
$$

which also works for the discussed degenerate case. The momentum shift $\boldsymbol{\delta}$ is along a (k-space-) nearest-neighbor plaquette and the normalization constant $\mathcal{N}_{\boldsymbol{\delta}}(\mathbf{k})$ may be neglected if phases of the complex numbers are directly obtained.

Once a self-consistent solution for the density matrix is obtained on a grid $\{\mathbf{k}\}$, the corresponding eigenvectors of the quadratic HF Hamiltonian $\left| u_n^{\mathrm{HF}}(\mathbf{k}) \right\rangle$ fully characterize the Slater determinant state. Let $\alpha$ denote all internal degrees of freedom, in our case spin $\sigma$, valley $\tau$ and $\mathbf{Q}$, and $d_{n,\mathbf{k}}^\dagger$ be the creation operators of the self-consistent HF quasiparticles, related to the original band fermions by $d_{n,\mathbf{k}}^\dagger = \sum_\alpha u_{n;\alpha}^{\mathrm{HF}}(\mathbf{k}) c_{\alpha,\mathbf{k}}^\dagger$. Then the HF ground state is given by

$$
\left| \Psi_0^{\mathrm{HF}} \right\rangle = \left( \prod_{\mathbf{k} \in \mathrm{RBZ}} \prod_n^{n_F} d_{n,\mathbf{k}}^\dagger \right) |0\rangle,
\tag{26}
$$

where $M = \left| \{\mathbf{k}\} \right| = N/4$ and $M \times n_F = N_e$. In the band-projected basis, the state is composed as

$$
\begin{aligned}
\left| \Psi_0^{\mathrm{HF}} \right\rangle &= \left( \prod_{\mathbf{k}} \prod_n^{n_F} \left[ \sum_\alpha u_{n;\alpha}^{\mathrm{HF}}(\mathbf{k}) c_{\alpha,\mathbf{k}}^\dagger \right] \right) |0\rangle = \left( \prod_{\mathbf{k}} \prod_n^{n_F} \left[ u_{n;\alpha_1}^{\mathrm{HF}}(\mathbf{k}) c_{\alpha_1,\mathbf{k}}^\dagger + \cdots + u_{n;\alpha_{N_\alpha}}^{\mathrm{HF}}(\mathbf{k}) c_{\alpha_{N_\alpha},\mathbf{k}}^\dagger \right] \right) |0\rangle \\
&= \left( \prod_{\mathbf{k}} \left\{ u_{1;\alpha_1}^{\mathrm{HF}}(\mathbf{k}) u_{2;\alpha_2}^{\mathrm{HF}}(\mathbf{k}) \cdots c_{\alpha_1,\mathbf{k}}^\dagger c_{\alpha_2,\mathbf{k}}^\dagger \cdots + u_{1;\alpha_2}^{\mathrm{HF}}(\mathbf{k}) u_{2;\alpha_1}^{\mathrm{HF}}(\mathbf{k}) \cdots c_{\alpha_2,\mathbf{k}}^\dagger c_{\alpha_1,\mathbf{k}}^\dagger \cdots + \cdots \right\} \right) |0\rangle, \quad (27)
\end{aligned}
$$

with a subset $A = \{\alpha_i\}_{i=1}^{n_F}$ sorted like $\alpha_1 < \alpha_2 < \cdots < \alpha_{n_F}$, giving $\binom{N_\alpha}{n_F}$ possibilities. Then

$$\left|\Psi_0^{\text{HF}}\right\rangle = \Big(\prod_{\mathbf{k}}\sum_A \underbrace{\Big[\sum_{\{\beta_n\}\in\pi(A)}(-1)^{\text{sgn}(\{\beta_n\})}\prod_n^{n_F} u_{n;\beta_n}^{\text{HF}}(\mathbf{k})\Big]}_{\xi_{\mathbf{k}}(A)}\prod_{\alpha\in A}c_{\alpha,\mathbf{k}}^\dagger\Big)|0\rangle = \Big(\prod_{\mathbf{k}}\sum_A \xi_{\mathbf{k}}(A)\prod_{\alpha\in A}c_{\alpha,\mathbf{k}}^\dagger\Big)|0\rangle$$

$$= \Big(\Big[\sum_{A^1}\xi_{\mathbf{k}^1}(A^1)\prod_{\alpha^1\in A^1}c_{\alpha^1,\mathbf{k}^1}^\dagger\Big]\Big[\sum_{A^2}\xi_{\mathbf{k}^2}(A^2)\prod_{\alpha^2\in A^2}c_{\alpha^2,\mathbf{k}^2}^\dagger\Big]\cdots\Big[\sum_{A^M}\xi_{\mathbf{k}^M}(A^M)\prod_{\alpha^M\in A^M}c_{\alpha^M,\mathbf{k}^M}^\dagger\Big]\Big)|0\rangle$$

$$= \Big(\sum_{A^1}\Big\{\xi_{\mathbf{k}^1}(A^1)\prod_{\alpha^1\in A^1}c_{\alpha^1,\mathbf{k}^1}^\dagger\Big[\sum_{A^2}\xi_{\mathbf{k}^2}(A^2)\prod_{\alpha^2\in A^2}c_{\alpha^2,\mathbf{k}^2}^\dagger\Big]\cdots\Big\}\Big)|0\rangle$$

$$= \Big(\sum_{A^1\cdots A^M}\prod_{j=1}^M\xi_{\mathbf{k}^j}(A^j)\prod_{\alpha^j\in A^j}c_{\alpha^j,\mathbf{k}^j}^\dagger\Big)|0\rangle \quad\text{, where }\left|\{A^1\cdots A^M\}\right| = \binom{N_\alpha}{n_F}^M,$$

$$= \Big(\sum_{\{A^j\}}\Big[\prod_{j=1}^M\xi_{\mathbf{k}^j}(A^j)\Big]\underbrace{\prod_{j=1}^M\prod_{\alpha^j\in A^j}c_{\alpha^j,\mathbf{k}^j}^\dagger}_{(-1)^{\text{sgn}(\{\mu_i\})}\prod_{\mu_i}c_{\mu_i}^\dagger}\Big)\quad\text{, with }\mu_i = (\alpha^j,\mathbf{k}^j),\ \mu_1 < \mu_2\cdots < \mu_{N_e},$$

$$= \Big(\sum_{\{A^j\}}\underbrace{(-1)^{\text{sgn}(\{\mu_i\})}\prod_{j=1}^M\xi_{\mathbf{k}^j}(A^j)}_{\Xi(\{A^j\})}\prod_{i=1}^{N_e}c_{\mu_i}^\dagger\Big)|0\rangle = \Big(\sum_{\{A^j\}}\Xi(\{A^j\})\prod_{i=1}^{N_e}c_{\mu_i}^\dagger\Big)|0\rangle. \tag{28}$$

Since the set $\{\mu_i\}$ is uniquely determined given a collection $\{A^j\}$ and an order of the single-particle orbitals $(\alpha,\mathbf{k})$, each summand acts on a separate Fock space basis state and the $\Xi(\{A^j\})$ are nothing but the coefficients of the decomposition in this basis. Furthermore, choosing the order $\mu_1 < \mu_2 < \cdots < \mu_{N_e}$ when applying the product of creation operators like $c_{\mu_1}^\dagger c_{\mu_2}^\dagger\cdots c_{\mu_{N_e}}^\dagger|0\rangle$ gives the corresponding Fock state $|0\cdots010\cdots010\cdots01\cdots\rangle$ without an additional sign. The index of the orbital to be occupied is given by $\mu_i = (\alpha,\mathbf{k})$, where $1\leqslant\mu_i\leqslant N_\alpha\times M$. The representation of the HF state given in Eq. (28) can be used to perform the decomposition in the ED spectrum and hence compute the weight of the Slater determinant in the ground state manifold. This, in turn, constitutes a quantitative measure for how close the ED ground state is to a product state.

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
