# Peer review of "Non-coplanar magnetism, topological density wave order and emergent symmetry at half-integer filling of moiré Chern bands"

_SciPost Physics, doi:SciPost Phys. 14, 040 (2023)_

## Round 1 · Referee Report · Anonymous (Referee 1) · 2022-9-12

Report

The authors performed numerical studies on twisted double-bilayer graphene and twisted mono-bilayer graphene at nu=1/2 and 7/2 to explain the recently observed Chern insulator in Nat. Phys. 18(1), 42 (2022). Their numerical results from both exact diagonalization and unrestricted Hartree-Fock unanimously show the energetically-favorable ground state of tetrahedral antiferromagnetism at small w_AA/w_AB, and three almost degenerate ferromagnetic states with an approximate emergent O(3) symmetry at large w_AA/w_AB. The result itself is very interesting from a simple mechanism, and it further naturally provide an explanation for the experiment. Therefore, I recommend the publication of this manuscript, which is a good example of satisfying the journal standard: a pertinent and timely study on a groundbreaking topic, a clear logic and complete story, rigorous calculationsfrom multiple aspects, and explicit appendices documenting all details for other physicists to reproduce.

Requested changes

I just have a few cosmetic suggestions for the authors to consider: 1. There are many quantities defined throughout the paper, such as eta, N_nu, etc. I understand that the definitions are scattered somewhere in the main text. However, I suggest the authors also state them when they appear in equations and figure labels for better readability. For example, in App C2. Page 25, I understand that the definitions of p_nu, m, Chi, phi are in Eq. 7-9 in the main text, however, they may also consider stating them when they appear in the appendix.

  1. Figure 4 seems too crowded, for example, the x-axis "1/N" of (b) almost touches the top axis of (d). In (c), it is better if N is stated in the caption. (Otherwise, I can only guess N by counting the dots in the first Brillouin zone with the assumption that these dots are also the samplings in momentum space.) In (d), the y-axis label is not explicitly defined, I guess G=0 indicates the M points in the first Brillouin zone.

  2. "MRL" in Eq. 16 and "RL" above Eq. 17 are not defined explicitly. I guess they are "multiple reciprocal lattices" and "reciprocal lattice"? It's better to define it explicitly to avoid potential confusion.

  3. Fig 2b has no y-axis label, but a legend. I think the legend is unnecessary here because there is just one line, and can be placed as the y-axis label.

  • validity: top
  • significance: top
  • originality: high
  • clarity: top
  • formatting: perfect
  • grammar: perfect

Author:  Patrick Wilhelm  on 2022-11-15  [id 3019]

(in reply to Report 1 on 2022-09-12)

We thank the referee for carefully reading and the helpful comments to improve the quality of the paper. 1. We agree that readability can be improved by restating the definitions of variables when they appear away from their original one, especially in the appendix, and updated the manuscript accordingly. 2. Furthermore, Figs. 3 – 5 were indeed too crowded and hence adjusted with bigger spacings of the subfigures. 3. The missing definition of "MRL" in Eq. 16 was added and the typo "RL"-> "MRL" is now fixed. 4. We agree that the legend in Fig. 2(b) is unnecessary and better replaced by an actual y-axis label.

---

## Round 1 · Referee Report · Anonymous (Referee 2) · 2022-11-7

Strengths

This paper reports a theoretical study of interaction effects in moire Chern bands pertinent to twisted double bilayer graphene or twisted monolayer-bilayer graphene. The methods used are exact diagonalization study of many body Hamiltonians supplemented with a Hartree-Fock calculation. The research is well motivated by experiments on both of these systems.

The results from the study reveal a few interesting features at half-integer filling of the moire bands: first, in addition to a valley polarization, the spin forms a non-collinear tetrahedral antiferromagnet pattern over a range of parameters - specifically for not too large ratio of the AA to AB hopping. When this ratio increases there is a phase transition to a spin ferromagnetic phase which has a coexisting charge ordering pattern with an approximate emergent O(3) symmetry. Both phases have a quantized electrical Hall conductivity.

The results are related to existing experiments on these systems.

Weaknesses

There is no major weakness in the study. As a matter of taste, I might have liked to see more discussion of the physics that drives the formation of the tetrahedral spin order, or the nature of the finite temperature phase structure of this phase. However the main Exact Diagonalization results do not strictly require such additional discussion, and it is acceptable for the authors to keep the presentation as it is.

The referencing could also be improved. The theory literature on nearly flat Chern bands in moire systems goes back to the early days of this field (2018 or 2019) would be useful to reference. Similarly the early experiments on ferromagnetism and quantum anomalous Hall states in twisted bilayer graphene and on ABC trilayer graphene would also be useful references.

Report

I am happy to recommend the paper for publication.
  • validity: top
  • significance: high
  • originality: good
  • clarity: high
  • formatting: excellent
  • grammar: excellent

Author:  Patrick Wilhelm  on 2022-11-15  [id 3020]

(in reply to Report 2 on 2022-11-07)

We thank the referee for the helpful comments and suggestions.
Indeed, the finite temperature behavior poses an interesting future research direction. However, the relatively large gap of a few meV suggest that for low-enough temperatures (<~10 K) the physics should be dominated by the T=0 K manifold of TAF and TCDW states. Furthermore, we added additional references to early theoretical and experimental works in the field, which gives better context to the relevance of our study.

---

## Round 2 · Author Response

In accordance with the helpful suggestions of the referees, we updated the original manuscript as follows: 1) We added an additional remark about early studies of moiré Chern bands, both theoretical and experimental, in order to give more context on the development of the field. 2) Fig. 2 (b) was updated with a y-axis label, replacing the single entry in the legend. 3) We increased the spacings of sub-figures in Fig. 3 – 5 for better readability. 4) A note was added to Fig. 4 – 5, clarifying the role of the G=0 superscript, as well as an explicit statement of the used system size N. 5) The definition of "MRL" in Eq. 16 was added and the typo "RL"->"MRL" above Eq. 17 was corrected. 6) We restated the definitions of various variables scattered throughout the text, especially p_nu, m, Chi, phi in App. C.2.

---

## Editorial Decision

published